# Representational Difference Explanations

Neehar Kondapaneni[1]    Oisin Mac Aodha[2]    Pietro Perona[1]

[1]Caltech    [2]University of Edinburgh

## Abstract

We propose a method for discovering and visualizing the differences between two learned representations, enabling more direct and interpretable model comparisons. We validate our method, which we call *Representational Differences Explanations* (`RDX`), by using it to compare models with known conceptual differences and demonstrate that it recovers meaningful distinctions where existing explainable AI (XAI) techniques fail. Applied to state-of-the-art models on challenging subsets of the ImageNet and iNaturalist datasets, `RDX` reveals both insightful representational differences and subtle patterns in the data. Although comparison is a cornerstone of scientific analysis, current tools in machine learning, namely post hoc XAI methods, struggle to support model comparison effectively. Our work addresses this gap by introducing an effective and explainable tool for contrasting model representations. `Project Page: RDX`

## 1  Introduction

In recent years, deep learning researchers and engineers have explored the costs and benefits of using larger datasets and more complex architectures. These changes can often lead to distinct models with different representations of the same data. An intuitive approach to understanding the effects of different architectures and training choices is to analyze the *representational differences* between models. Dictionary learning (DL)-based explainable AI (XAI) methods, like sparse autoencoders (SAEs) and non-negative matrix factorization (NMF), are powerful tools for analyzing model representations that surface model *concepts*, i.e., semantically meaningful sub-components of the input data [9, 15, 26, 54]. These approaches are formulated as a dictionary learning problem [13] such that model representations are decomposed into a linear combination of learned concept vectors. Concept vectors are then converted into human-friendly explanations by selecting a subset of input items (e.g., a set of images) that maximally align with the vector. These explanations have been shown to help users better understand models [8, 15, 26, 54]. However, when adapting existing DL-based XAI methods for comparing models with known differences, we find that they often generate explanations that are unrelated to the known difference between models.

We identify three issues with existing DL-based XAI methods that limit their power of analysis, especially when *comparing* representations. First, when representational differences are relatively small, concepts from different models tend to overlap and it is thus difficult to spot differences. Second, we observe that existing methods explain concepts by sampling and visualizing items with the largest activations [9, 15, 17, 26, 62], which tend to be the ones that are easiest to interpret, and miss more nuanced differences, thus offering *incomplete* explanations. Finally, to understand subtle differences between models, users need to consider the weighted sum of several concepts *via their incomplete explanations* which can be difficult due to the imprecise nature of the task.

39th Conference on Neural Information Processing Systems (NeurIPS 2025).

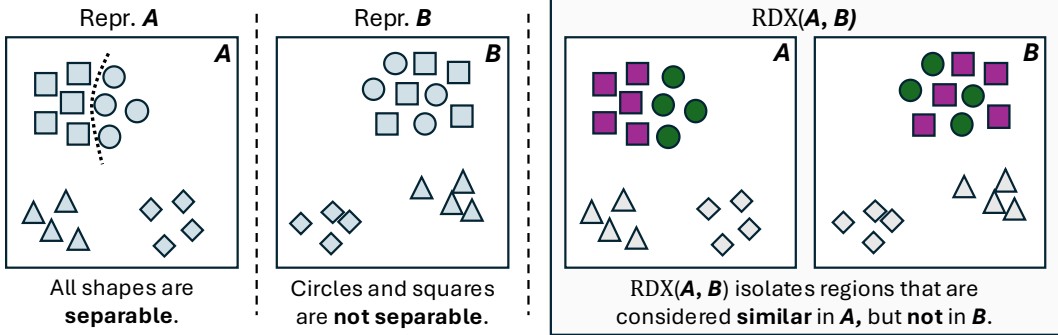

Figure 1: **Intuition behind our method.** *Representational Difference Explanations* (RDX) aim to highlight the substantive differences between two representations (e.g., *A* and *B*, which are the embedding matrices produced by two different models for the same set of data). Here *A* supports discrimination between circles and squares, whereas *B* does not. Clustering the two representations independently would not reveal the square/circle sub-structure unique to *A*. By "subtracting" *B* from *A*, RDX reveals which items are considered similar in *A*, but not in *B*. RDX isolates *differences*, and ignores data that may be equally well grouped in both representations, such as the triangles and diamonds.

We propose a new XAI method, named *Representational Difference Explanations* (RDX) for explaining model differences. Rather than focusing on one representation at a time, RDX compares two representations against each other to isolate the *differences* between them (Fig. 1). Additionally, unlike DL-methods which generate *incomplete* explanations for concept vectors, we take the perspective that a concept and its explanation are *one and the same thing*. To achieve this, we enforce that each concept is defined by a small set of semantically related samples from a dataset.

We make the following key contributions: (1) A new method, RDX, that identifies explainable differences between model representations. (2) A new metric to measure the effectiveness of such representational difference explanations. (3) Experiments comparing RDX against baseline methods for explaining model differences, demonstrating its superiority.

## 2 Related Work

**Explainability.** Explainable AI methods for computer vision attempt to generate explanations to help users understand model behavior. There are two broad classes of methods: *local methods* [4, 37, 55, 59] attribute regions of an image to a model's decision and often take the form of heatmaps, while *global methods* [18, 26, 72] generate a global explanation (e.g., a grid of images or image regions) that represent a visual concept that is learned by a model. Visual concepts are often represented by a set of images that are considered similar due to a shared visual feature. For example, the visual concept of 'red circle' maybe represented by a set of images that all contain red circles. Visual concepts can either be defined by a user-selected set of similar input images [26] or be discovered directly from the model by grouping images the model considers similar [13]. These visual concepts are used to help users achieve a more general understanding of model behavior [8, 15, 26, 28, 54]. For example, they can reveal that a model has learned to use water as a cue for detecting a certain species of waterbird. Local and global methods can also be combined to provide detailed explanations that describe both the concepts and the image regions used by a model when making a decision [1, 15, 29, 58, 62].

**Representational Similarity.** Representational similarity methods [21, 22, 30, 35, 50] aim to quantify the similarity between network representations. These methods operate by passing the same set of items through two models to generate two embedding matrices. These embedding matrices are then compared, resulting in a single value that quantifies their degree of similarity. While these approaches can provide useful, coarse-grained insights [34, 40, 42, 46, 51, 69, 73], they do not help with understanding fine-grained model differences. Recently, several methods have been proposed which compare networks through interpretable

concepts [28, 58, 62, 65]. RSVC [28], vision SAEs [58], and NLMCD [65] extract concepts independently for each model and match them in a subsequent step, resulting in partially overlapping concepts that can make interpretation challenging. USAEs [62] employ "universal" sparse autoencoders that must be trained for each new model and dataset to learn a common representational space across several models. This training step makes generalization to different models or datasets challenging. Additionally, none of these methods are designed to specifically seek out differences, although differences may be detected as a byproduct of their approaches. In contrast, our approach uses information from both representations simultaneously to discover differences between them. It requires no training, making it easy to apply to new models and bespoke datasets.

**Comparing Graphs.** Our approach is related to graph comparison methods. Many methods have been developed for comparing graphs, including methods for matching the largest common subgraphs [5], detecting anomalies [45], grouping network types [2], and measuring the similarity between graphs via kernels [66]. The majority of existing methods are concerned with developing specialized strategies for comparing very large web-scale graphs with mismatched nodes. In addition, these approaches aim to quantify network similarity with a score rather than to visualize and understand qualitative differences. In contrast, our goal is to provide fine-grained, qualitative understanding of the differences between two "graphs" that have the same nodes, but different edge weights. While some approaches [3, 48] have been developed to visualize differences between graphs, these methods focus on highlighting the addition and removal of nodes. Most relevant to our work is DiSC [57], a modification of the spectral clustering algorithm. DiSC addresses a setting in which there are two experimental conditions, where the same types of measurements are taken in both conditions. Given this shared feature space, it seeks out *features* that cluster together in one condition, but not in the other. This paradigm is relevant for biological experiments, in which genes may co-activate in certain experimental conditions. Our approach differs from DiSC in two key ways: (1) Different neural networks do not have a shared feature space, therefore we focus on discovering differential clusters of *inputs*, not features. (2) We construct an affinity matrix emphasizing the *difference* between representations. This makes our approach more flexible than DiSC, since it can be used with any clustering algorithm.

## 3 Method

We propose a method, RDX, to explain the differences between two models via concepts by identifying inputs that *only* one of the two models considers to be semantically related. To do so, we construct an affinity matrix that assigns high affinity to pairs of inputs that are similar according to representation $\boldsymbol{A}$, but dissimilar according to representation $\boldsymbol{B}$. We cluster this affinity matrix to reveal distinctive similarity structures in $\boldsymbol{A}$. At a high-level, RDX performs the following steps: (1) compute the pairwise distances between inputs in $\boldsymbol{A}$ and $\boldsymbol{B}$ to build distance matrices, $\boldsymbol{D}_A$ and $\boldsymbol{D}_B$, (2) compute the *normalized* difference between the matrices to form *difference* matrices $\boldsymbol{G}_{A,B}$ and $\boldsymbol{G}_{B,A}$, and (3) use the difference matrices to sample difference explanations, i.e., explanations that reveal where the two representations disagree. Intuitively, negative edges in $\boldsymbol{G}_{A,B}$ indicate that the corresponding pair of inputs were closer together in $\boldsymbol{A}$ than they were in $\boldsymbol{B}$.

As input, we have $n$ data items from which we compute two embedding matrices obtained from two different models, $\boldsymbol{A} \in \mathbb{R}^{n \times d_A}$ and $\boldsymbol{B} \in \mathbb{R}^{n \times d_B}$, where $d_A$ and $d_B$ are the embedding dimensions for models $A$ and $B$ respectively. $\boldsymbol{A}$ and $\boldsymbol{B}$ contain embeddings over the same set of inputs, where each row corresponds to the same input item, i.e., each row is an embedding vector. We refer to the $i^{th}$ embedding vector in $\boldsymbol{A}$ as $\boldsymbol{a}_i$. We consider several options for each step of RDX and provide details for the best choices in the following sections. Additional model variants are described in Appendix B.

### 3.1 Computing Normalized Distances

To contrast representations using their distance matrices, the distances must be comparable.

**Neighborhood Distances.** We compute the pairwise Euclidean distance matrices, $\boldsymbol{D}_A \in R^{n \times n}$ and $\boldsymbol{D}_B \in R^{n \times n}$, for each embedding matrix separately. For each entry $\boldsymbol{a}_i$ in a given embedding matrix, we rank all other entries by their distance to $\boldsymbol{a}_i$. This rank is used as the

scale-invariant nearest neighbor distance between $\boldsymbol{a}_i$ and $\boldsymbol{a}_j$. Thus, distances are integers between 1 and $n$. We refer to the outputs as the normalized distance matrices $\bar{\boldsymbol{D}}_A$ and $\bar{\boldsymbol{D}}_B$.

## 3.2 Constructing Difference Matrices

Given $\bar{\boldsymbol{D}}_A$ and $\bar{\boldsymbol{D}}_B$, we develop a method for comparing the normalized distances that emphasizes instances where either model considers two inputs similar. The method is asymmetric. Here we present it in one direction.

**Locally Biased Difference Function.**

Consider two pairs of embeddings with indices $i, j$ and $i, k$. Suppose $\bar{\boldsymbol{D}}_A^{ij} = 500$ and $\bar{\boldsymbol{D}}_B^{ij} = 600$ in the first pair, and $\bar{\boldsymbol{D}}_A^{ik} = 1$ and $\bar{\boldsymbol{D}}_B^{ik} = 101$ in the second. Comparing the distances across the representations $(500 - 600, 1 - 101)$ results in the same amount of change $(-100, -100)$, but a change from a distance of 1 to a distance of 101 suggests a more important conceptual difference. To address this issue, we propose a locally-biased difference function (Fig. A12):

$$\boldsymbol{G}_{A,B}^{ij} = \tanh(\gamma \cdot (\bar{\boldsymbol{D}}_A^{ij} - \bar{\boldsymbol{D}}_B^{ij})/(\min(\bar{\boldsymbol{D}}_A^{ij}, \bar{\boldsymbol{D}}_B^{ij})). \tag{1}$$

By dividing by the minimum distance across both representations, this function prioritizes differences in embedding distances in which either representation considers the embeddings to be similar. This ensures that large differences in distant embeddings are ignored, but large differences in nearby embeddings are emphasized. To avoid exponential growth in our difference function when distances are small, we apply a tanh function to normalize the outputs, with $\gamma$ controlling how quickly the function saturates. Given an item indexed by $i$, this function will output negative values when the distance between $i$ and $j$ is smaller in $\boldsymbol{A}$ than in $\boldsymbol{B}$. Thus, negative matrix entries denote items that are closer in $\boldsymbol{A}$ than in $\boldsymbol{B}$.

## 3.3 Sampling Difference Explanation Grids

The next step is to communicate representational differences to the user. Sets of images, presented as an image grid of 9-25 images, have been used to communicate concepts for visual data [15, 26]. We aim to sample $m$ sets of images (i.e., image grids) from a difference matrix. Each set of images should contain images that are considered similar by only $\boldsymbol{A}$, i.e., indices that have pairwise negative matrix entries in $\boldsymbol{G}_{A,B}^{ij}$. We refer to this set of images as a *difference explanation* ($E$) that defines a concept unique to one model and we refer to the collection of difference explanations as $\mathcal{E}^{\boldsymbol{A}}$. If we consider the difference matrix as the adjacency matrix of a graph, we are essentially looking for a subgraph of size $|E|$ with large negative values on all edges. There are many options for sampling subgraphs, we consider a direct sampling (see Appendix B.5) and a spectral clustering based approach.

**Spectral Clustering.** We convert $\boldsymbol{G}_{A,B}$ into an affinity matrix: $\boldsymbol{F}_{A,B} = \exp(-\beta \cdot \boldsymbol{G}_{A,B})$. To ensure the affinity matrix is symmetric, we average it with its transpose. From this affinity matrix, we seek to sample a set of $m$ difference explanation grids $\mathcal{E}^{\boldsymbol{A}}$. Given an affinity matrix, spectral clustering solves a relaxed version of the normalized cut problem [67]. Normalized cuts (N-Cut) seek out a partition of a graph that minimizes the sum of the cut edges, while balancing the size of the partition [56]. Since, edges in $\boldsymbol{F}_{A,B}$ are large when inputs are closer in $\boldsymbol{A}$ than they are in $\boldsymbol{B}$, spectral clustering is biased to finding partitions in which inputs are close together in $\boldsymbol{A}$, but far apart in $\boldsymbol{B}$. In practice, when both representations have a similar structure, edges in that structure will have an affinity close to 1, since the difference is near 0. To discard these regions, we generate $m + 1$ clusters and discard the cluster with lowest mean affinity as it contains regions we are uninterested in. Spectral clusters can contain too many inputs to be visualized all at once. To convert each cluster into an explanation grid ($E$), we define the k-neighborhood affinity (KNA). For each image in the spectral cluster, the KNA is the sum of the edges between that image and its k-nearest neighbors (from within the cluster). Recall that larger affinity edges indicate more disagreement about the similarity between two images, thus we select the image and neighbors corresponding to the max KNA (see pseudo-code in Appendix B.1).

## 3.4 Representational Alignment

When models have significant representational differences, it is possible that these differences could be mitigated by aligning the representations. For example, both models may be the same up to some (e.g., linear) transformation. In these settings, it can be useful to first maximize the alignment between models before exploring the representational differences, since this can reveal fundamental differences between the models. To align the representation $\boldsymbol{A} \in \mathbb{R}^{n \times d_A}$ to $\boldsymbol{B} \in \mathbb{R}^{n \times d_B}$, we learn a transformation matrix $\boldsymbol{M}_{AB} \in \mathbb{R}^{d_A \times d_A}$ that minimizes the centered kernel alignment (CKA) loss [53] between the transformed $\boldsymbol{A}$ and $\boldsymbol{B}$:

$$\boldsymbol{M}_{AB}^* = \arg\min_{\boldsymbol{M}} 1 - \mathrm{CKA}(\boldsymbol{AM}, \boldsymbol{B}), \qquad (2)$$

where $\mathrm{CKA}(\cdot, \cdot)$ represents the linear CKA similarity. We denote $\boldsymbol{A}$ aligned to $\boldsymbol{B}$ as $\boldsymbol{A}'$. See Appendix C.2 for training details.

## 3.5 Evaluating Explanations

All explanation methods in this work produce a set of explanation grids for both representations, i.e., recall that $\mathcal{E}^{\boldsymbol{A}}$ is the set of explanation grids for $\boldsymbol{A}$. The goal of a difference explanation grid is to identify sets of items that are closer together in one representation than they are in the other. We develop a metric, the binary success rate (BSR), to evaluate whether a method has succeeded at this task. $\bar{\boldsymbol{D}}_A^{ij}$ represents the distance between two items in representation $\boldsymbol{A}$ and $\bar{\boldsymbol{D}}_B^{ij}$ represents the same for representation $\boldsymbol{B}$. We measure how frequently the

**Algorithm 1: Evaluation of Explanations on Representation A.**

1: **Input:** Grids $\mathcal{E}^{\boldsymbol{A}} = \{E_1, \ldots, E_k\}$,
   and distances $\bar{\boldsymbol{D}}_A, \bar{\boldsymbol{D}}_B$
2: `BSR = 0`
3: **for** each grid $E \in \mathcal{E}^{\boldsymbol{A}}$ **do**
4:    **for** each pair $(i, j) \in E$, $i \neq j$ **do**
5:       **if** $\bar{\boldsymbol{D}}_A^{ij} < \bar{\boldsymbol{D}}_B^{ij}$ **then**
6:          `BSR += 1`
7:       **end if**
8:    **end for**
9: **end for**

distance for a pair of items from an explanation grid is smaller in $\boldsymbol{A}$ than in $\boldsymbol{B}$. In the pseudo-code above, we provide the algorithm for computing $\mathrm{BSR}(\mathcal{E}^{\boldsymbol{A}})$. In addition to the BSR metric, we compute three metrics from SemanticLens [12]: Redundancy (modified), Clarity, and Polysemanticity. We present details for computing these metrics in Appendix B.2.

## 4 Results

We conduct three experiments to evaluate the effectiveness of RDX. In Sec. 4.2, we compare two simple representations with subtle differences to show that existing XAI methods fail to explain these differences. In Sec. 4.3, we show that these trends continue to hold in more realistic settings. Through modifications of various models, we manipulate representations to have "known" differences. We then stage comparisons and assess whether existing XAI methods are able to recover these differences. In Sec. 4.4, we use RDX to compare models with unknown differences and find that it can reveal novel insights about models and datasets.

## 4.1 Implementation Details

We use several models in our evaluation. Unless specified otherwise, for our modified MNIST experiments, we use a 2-layer convolutional network with an output dimension of 64. We also train a post-hoc concept bottleneck model (PCBM) [70] with a ResNet-18 [20] backbone on the CUB dataset [68] using a standard training procedure [70]. Finally, we conduct experiments using several models that are available from the timm library [64]: DINO [6] vs. DINOv2 [43] and CLIP [49] vs. CLIP-iNat (i.e., a CLIP model fine-tuned on data from the iNaturalist platform [24]). In these experiments, models are compared on subsets of images from 2-4 commonly confused classes in ImageNet [10] or iNaturalist [24]. More training details are provided in Appendix C.1. We compare our approach to several DL for XAI baselines: top-k sparse auto-encoders (TKSAE) [17, 38], sparse auto-encoders (SAE) [41], non-negative matrix factorization (NMF) [33], principal component analysis (PCA) [16], and KMeans [36]. We use convex non-negative matrix factorization (CNMF) [11] if the activations of the last layer contains negative values. We also compare to a method explicitly designed for model comparison, non-linear multi-dimensional concept discovery (NLMCD) [65]. We provide

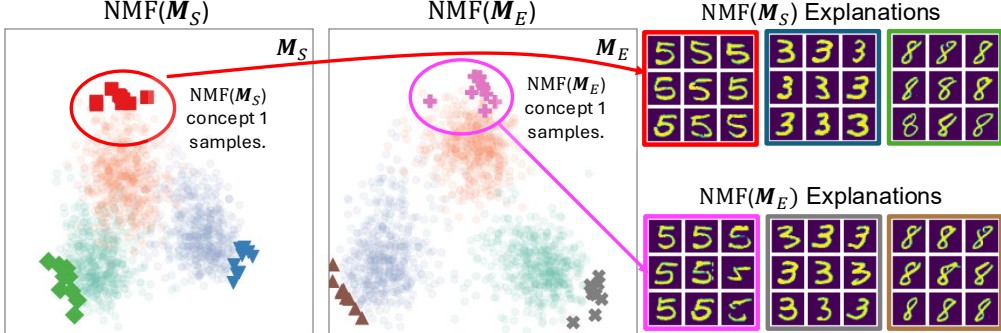

**(A)** NMF explanations are indistinguishable despite significant differences in representation.

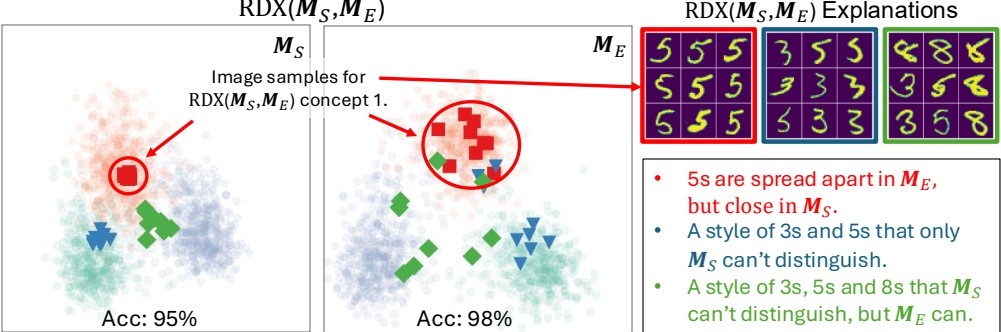

**(B)** RDX highlights differences between $M_S$ and $M_E$, by identifying concepts specific to $M_S$.

Figure 2: **Comparing RDX to NMF.** We train a small CNN on a modified MNIST dataset that only contains images of the digits 3, 5, and 8. We compare a strong model checkpoint representation ($M_S$, 95% accuracy) with a final 'expert' model representation ($M_E$, 98% accuracy). The left and middle columns show PCA projections of the $M_S$ and $M_E$ representations, respectively. The transparent colors indicate classes in the dataset: 3 (light-blue), 5 (light-orange), and 8 (light-green). The right most columns visualize the images selected by the explanation methods. We extract three concepts for each method. **(A)** We generate explanations using NMF [15] with maximum sampling [13, 15, 28] for $M_S$ and $M_E$. Bold colored points on the PCA plots indicate the location of the sampled images seen in the right-most column. We find that NMF is unable to reveal any representational difference between $M_S$ and $M_E$ because it produces indistinguishable explanations for both models. **(B)** In contrast, RDX discovers concepts unique to $M_S$ by identifying images that are more similar in $M_S$ than in $M_E$. The sampled points are overlaid on both models' representations and show tight clusters in $M_S$ that contrast with diffuse points in $M_E$. The right column shows the corresponding explanations, highlighting how model representations differ.

details on baseline methods in Appendix C.3 and details for RDX are provided in Appendix C.4. We conduct ablations in Appendix B.

## 4.2 Dictionary Learning Fails to Reveal Differences in Similar Representations

Dictionary learning (DL) approaches for XAI are commonly used to discover and explain vision models [15, 58, 62, 72]. We hypothesize that explanation grids sampled from DL concepts are not helpful for describing *differences* between similar representations, even if the representations contain behaviorally significant differences. To test this, we train a 2-layer convolutional network with an output dimension of 8 on a modified MNIST dataset that contains only images for the digits 3, 5, and 8. We compare a checkpoint from epoch 1 with strong performance (95% accuracy) to the final, 'expert' checkpoint at epoch 5 (98%). We refer to these checkpoints as $M_S$ (strong model) and $M_E$ (expert model). We conduct this experiment to assess if an XAI method can discover subtle differences between two models.

A good difference explanation should reveal the concepts that explain why $M_S$ under-performs $M_E$ by 3%. In Fig. 2 A we show that NMF with maximum sampling generates effectively the same explanation grid for both representations. This is because NMF has learned

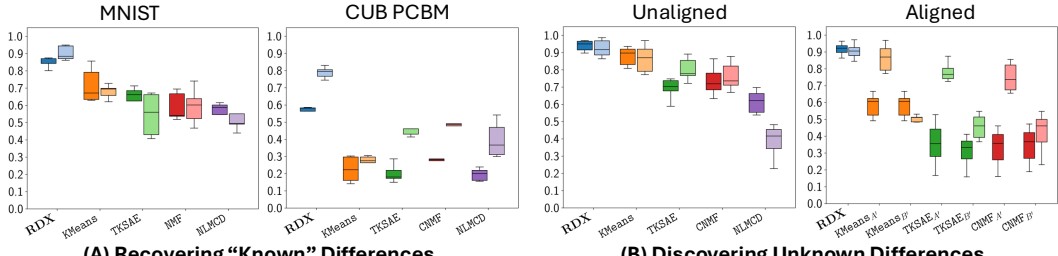

| MNIST | CUB PCBM | Unaligned | Aligned |
| --- | --- | --- | --- |

**(A) Recovering "Known" Differences** · **(B) Discovering Unknown Differences**

Figure 3: **Binary success rate evaluation of XAI methods.** For each XAI method, we compute the binary success rate (BSR) (Sec. 3.5) on all difference experiments, where higher is better. We use neighborhood distances to measure BSR (Sec. 3.1). Each method (x-axis) is assigned a different color, we show $\text{BSR}(\mathcal{E}^A)$ (darker box) and $\text{BSR}(\mathcal{E}^B)$ (lighter box). **(A)** We show results on the MNIST and CUB PCBM experiments (Sec. 4.3), in which we modify a representation and test if RDX can help identify the modification. **(B)** We show results when comparing large vision models with unknown differences (Sec. 4.4). We compare recovering differences without (left) and with (right) an initial alignment step (Sec. 3.4). In all cases, our RDX approach consistently outperforms the dictionary learning baselines. A complete set of results is available in Table A3.

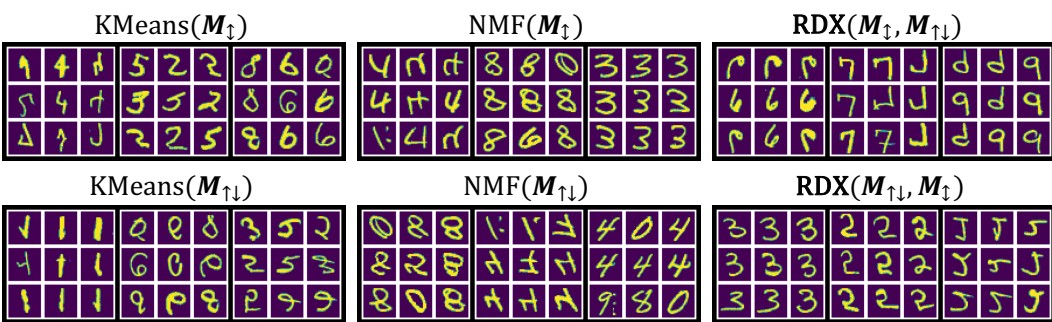

Figure 4: **Recovering vertical flip modifications on MNIST.** We visualize explanations produced by three XAI methods, RDX, KMeans, and NMF, to compare models $M_\updownarrow$ and $M_{\uparrow\downarrow}$. Both models are trained on a dataset with vertically flipped and normal images. $M_\updownarrow$ is trained to associate the original label to flipped digits and $M_{\uparrow\downarrow}$ is trained to predict a new set of labels for flipped digits. We expect $M_\updownarrow$ to mix flipped and unflipped digits while $M_{\uparrow\downarrow}$ should separate them. We generate three explanations for each method. **(Left, Middle)** KMeans and NMF generate explanations that are difficult to understand. **(Right)** RDX captures the expected difference. $\text{RDX}(M_\updownarrow, M_{\uparrow\downarrow})$ reveals that $M_\updownarrow$ represents flipped and unflipped 6s, 7s, and 9s closer together than in $M_{\uparrow\downarrow}$. $\text{RDX}(M_{\uparrow\downarrow}, M_\updownarrow)$ shows that $M_{\uparrow\downarrow}$ has clean clusters of 3s, flipped 5s, and flipped 2s without any mixing.

highly similar concepts for both representations, and the representational differences are captured in smaller and noisier concept coefficients for images that $M_S$ is less certain about. Maximum sampling selects the images with the *largest* coefficients, meaning these images are not sampled when visualizing concepts (Fig. A2 A). In Fig. A1, we show that SAE and KMeans also fail to explain representational differences. An alternative approach to understanding differences could be to inspect individual images of interest and try to understand them through their concepts. In Fig. A2 B, we show the difficulty of reasoning about an image via a weighted combination of concept explanations. In contrast, RDX concepts are equivalent to their explanation grid and are sampled from regions of differences. This ensures that RDX explanations select images considered similar by $M_S$ that $M_E$ does not consider similar. In Fig. 2 B, we can see that $M_S$ is confused by certain styles of 3s, 5s, and 8s that look similar when compared to $M_E$. In Fig. A1 C we visualize the reverse direction for RDX and find that $M_E$ contains clusters of challenging 3s and 5s, that are confused by $M_S$. Finally, in Appendix A.1.3 we discuss if perfectly monosemantic DL concepts would solve these issues. We argue that monosemanticity is likely infeasible when trying to compare representational differences and that, even if achieved, it cannot solve the challenge of interpreting weighted combinations of explanations.

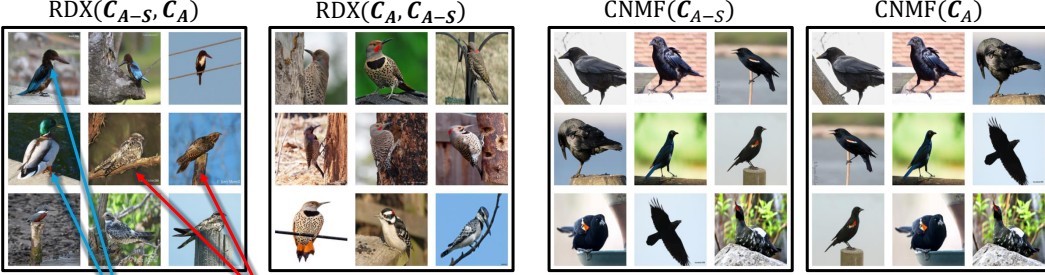

| RDX($\boldsymbol{C}_{A-S}$, $\boldsymbol{C}_A$) | RDX($\boldsymbol{C}_A$, $\boldsymbol{C}_{A-S}$) | CNMF($\boldsymbol{C}_{A-S}$) | CNMF($\boldsymbol{C}_A$) |

Mix of non-spotted and spotted wings in explanation for $\boldsymbol{C}_{A-S}$. In contrast, only spotted wings in $\boldsymbol{C}_A$ explanation.

Indistinguishable explanations for both models that are unrelated to wing spotted-ness.

Figure 5: **Recovering the "Spotted Wing" concept in CUB.** We train a post-hoc concept bottleneck model on the CUB dataset. For each image, we use the predicted concept scores as the image's embedding vector (i.e., representation). Here we compare a model using the complete concept representation ($\boldsymbol{C}_A$) with a model representation *without* the spotted wing concept ($\boldsymbol{C}_{A-S}$). We visualize one of five generated explanations for each model using RDX and CNMF. We observe that RDX's explanation focuses on the spotted wing concept. It shows us that only $\boldsymbol{C}_{A-S}$ mixes images with and without spotted wings. In contrast, the CNMF explanations for each model are both unrelated to the spotted wing concept and indistinguishable from each other, since the representations are highly similar and CNMF discovers nearly the same concepts in both. See Fig. A4 for all five explanations.

## 4.3 RDX Recovers "Known" Differences

Here we evaluate different XAI approaches by comparing MNIST trained models that have a modified training procedure. We select modifications such that we can have strong expectations on the differences between the learned representations (see Table A8 for a full list). For example, we trained two models on a MNIST dataset with vertically flipped digits, where $M_{\updownarrow}$ was trained with the same labels for both normal and flipped digits and $M_{\uparrow\downarrow}$ was given new labels for flipped digits. We expect that only $M_{\updownarrow}$ will mix flipped and unflipped digits. In Fig. 4, we visualize the outputs of three XAI methods for comparing $M_{\updownarrow}$ and $M_{\uparrow\downarrow}$. We clearly see that RDX's explanations focus on the actual expected difference. It shows that $M_{\updownarrow}$ considers flipped and normal digits as being more similar than $M_{\uparrow\downarrow}$. In contrast, KMeans and NMF result in unfocused and seemingly random explanations. In Fig. 3 A (left) we can see that this trend is consistent as all baseline methods have a lower BSR than RDX.

To explore differences between models trained on more complex images, we use a post-hoc concept bottleneck model (PCBM) trained on the CUB bird species dataset (Appendix C.1). The CUB PCBM ($\boldsymbol{C}_A$) predicts a score for 112 human-defined concepts, these concepts are then used to make species classification decisions, where we treat the concept predictions as a feature vector for an image. In each comparison, we remove a single concept from the feature vectors and compare the representations. The list of eliminated concepts used in this experiment can be found in Table A8. In Fig. 3 A (right) we report the BSR score for each method for this experiment. We find that RDX performs better than the baselines, especially for difference explanations showing concepts unique to the complete representation $\boldsymbol{C}_A$. In Fig. 5, we visualize the outputs of RDX and CNMF when comparing a model without the spotted wing concept ($\boldsymbol{C}_{A-S}$) against $\boldsymbol{C}_A$. As expected, we find that difference explanations show that $\boldsymbol{C}_{A-S}$ mixes images with and without spots, whereas, $\boldsymbol{C}_A$ is much better at grouping images with spotted wings. In contrast, we show that CNMF can result in both unrelated and indistinguishable explanations. We show more examples in Appendix A.2. Taken together, these results indicate that RDX is capable of revealing how changes in both training and fine-grained concepts can affect a model's representation.

## 4.4 RDX Discovers "Unknown" Differences

In our final experiment, we test the effectiveness of RDX for knowledge discovery by applying it to two models with unknown differences. We compare DINO with DINOv2 on four groups of ImageNet classes. We also compare CLIP against an iNaturalist fine-tuned CLIP (CLIP-iNat) model on three groups of different species. We conduct all of the knowledge discovery experiments with and without alignment. We align one model at a time, resulting

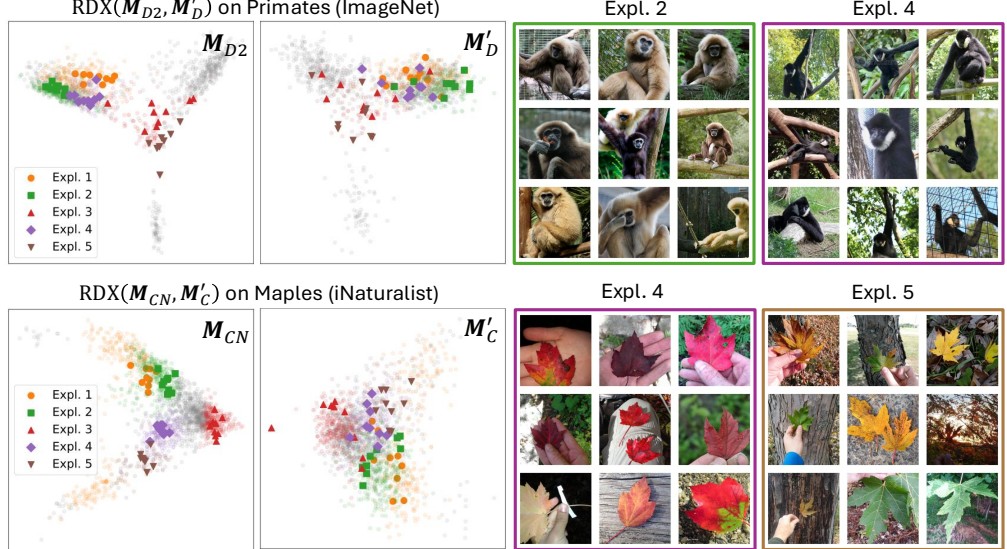

Figure 6: **Discovering unknown differences.** We use RDX to generate difference explanations for representations with unknown differences. We visualize two comparisons with alignment. In both comparisons, we visualize the shared structure (gray), spectral cluster membership (light colors), and selected samples for explanations (dark colors) on PCA projections of the representations. We can see that the selected indices are grouped compactly in the left PCA plot, but are spread apart in the right one. $(\text{RDX}(M_{D2}, M'_D))$ **on Primates**. We discover unique concepts in DINOv2 for commonly confused primates in ImageNet. In the PCA plot, we see that the green (Expl. 2) and purple (Expl. 4) explanations are cleanly separated in $M_{D2}$, but mixed in $M'_D$. The explanations show that only DINOv2 has unique concepts for tan-colored gibbons and for gibbons with white chin fur. $(\text{RDX}(M_{CN}, M'_C))$ **on Maples**. We compare the representations of CLIP-iNat and CLIP on four types of maple trees from iNaturalist. We see that CLIP-iNat contains a unique concept for fall-colored Red Maple leaves (Expl. 4) and a second concept that mixes fall-colored and green Silver Maple leaves (Expl. 5). Further analysis is provided in Sec. 4.4.

in twice the number of comparisons for baseline methods. We can see in Fig. 3 B that RDX outperforms all baseline methods in discovering representational differences. Additionally, we see that alignment makes it more challenging to discover differences for the baseline methods, but RDX maintains good performance in both settings. In Fig. 6 $(\text{RDX}(M_{D2}, M'_D))$, we visualize difference explanations by comparing DINOv2 to DINO using images from three primate classes from ImageNet. We find that DINOv2 does a better job at organizing two types of gibbons with different visual characteristics, suggesting that it would be more capable than DINO at fine-grained classification. In Fig. 6 $(\text{RDX}(M_{CN}, M'_C))$, we visualize fine-grained difference explanations on species of maple trees. We find that *only* CLIP-iNat contains well-separated concepts for two different species of maple, despite both clusters sharing a secondary characteristic of leaves with fall-colors. While CLIP does not mix the images from these concepts, we see that it does not group them as tightly, suggesting it may be organizing images using a different characteristic like color. We apply RDX to several more examples in Appendix A.3 and use a vision language model to assist in the analysis.

## 4.5 Additional Metrics and Concept Consistency

Finally, we present detailed results for each comparison using the BSR and SemanticLens [12] metrics in Appendix A.4. As expected, we find that RDX is the best performer on the BSR (Table A3) and Redundancy (Table A4) metrics. It is comparable to other methods on the Clarity (Table A5) and Polysemanticity (Table A6) metrics. We also conduct an analysis of concept consistency under in-distribution dataset variations in Appendix A.5 and find that RDX generates reasonably consistent concepts Fig. A11.

# 5 Discussion

Here we discuss some of the limitations of `RDX` and our analysis.

## 5.1 Limitations

**Compute.** Computing and storing the full pairwise distance matrix requires $\mathcal{O}(n^2)$ memory, which may become impractical for large $n$. In this work, we are able to apply our method to at least 5000 data points but we have not explored larger values of $n$.

**Concept Definition.** While our decision to define concepts via a grid of images is helpful for users, it does not allow us to communicate concepts like "roundness" that may vary linearly in response to objects with increasing roundness. Instead, concepts like "roundness" would be discretized into sub-concepts that can be communicated by an explanation grid.

**BSR agreement with human-interpretability.** In Fig. A15, we find that two methods can have the same `BSR`, but have significant differences in what they focus on in their explanations. Thus, we propose that `BSR` should not be directly optimized for, but should instead be a proxy metric, and that qualitative results should always be used to support `BSR` scores.

**Alignment.** Comparing after alignment is more likely to result in detecting fundamental representational differences. However, it is possible that there are two different aligned representations that result in the same alignment training loss. This would lead to different, but equally valid explanations and would require users to reason about feature correlations.

**Dependence on Euclidean distances.** `RDX` relies on Euclidean distances which could cause issues for accurate concept discovery. Specifically, given two datapoints, it is possible that the Euclidean distance between the points is small, but the points are quite far apart along the data manifold in the representation. `RDX` does not explicitly handle this issue, but there are two main reasons it is likely not a major concern. We discuss these reasons in Appendix A.6.

**Breadth.** Our approach works on any representation, but we focus on vision models in our experiments. Future work should explore if this approach can be useful when comparing text and multi-modal representations.

**Utility.** We find that `RDX` explanations are useful for identifying representational differences, but more work needs to be done to link these representational differences to performance differences on specific tasks such as classification. In Appendix A.2 (Maples), we see only some `RDX` explanations align with differences in classification. `RDX` is unsupervised in that it only requires two sets of representations as input, but in future work it would be interesting to explore incorporating classifier information into `RDX` explanations.

## 5.2 Societal Impact

We do not anticipate any specific ethical or usage concerns with the method proposed in this work. We propose a method for model comparison which we hope will improve our understanding of model representations. Deeper insight can lead to better detection of bias, better understanding of methods, and discovery of new knowledge about our datasets that may be beneficial for society. However, better understanding can also amplify negative usages of AI.

# 6 Conclusions

As models become larger and more powerful they are encoding more and more distinct concepts. As a result, describing and understanding these discovered concepts is an important question in XAI. In this work, we posit that comparing representations allows us to filter away common structure and reveal concepts that may be more interesting to the user. To achieve this, we introduced `RDX`, a new approach for isolating the *differences* between two representations. `RDX` requires no training, can be applied to any model that generates an embedding for an input, and is a general framework that can easily be modified with different choices for its intermediate steps. In our experiments we show that `RDX` is able to recover known model differences, and is also able to surface interesting unknown differences. The resulting observations can provide insight into model differences and the training data used by different models. The next step is testing `RDX` in real-life applications to see if it can be used to help experts, such as radiologists or ecologists, discover new concepts in their models and datasets.

**Acknowledgements.** We thank Atharva Sehgal, Rogério Guimarães, and Michael Hobley for providing feedback on the work. OMA was supported by a Royal Society Research Grant. NK and PP were supported by the Resnick Sustainability Institute.

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

# Appendix

## A   Additional Results

### A.1   Additional Results on MNIST-[3,5,8]

We train a small convolutional network on a modified MNIST dataset containing only images for digits 3, 5 and 8. We compare two checkpoints from training at epoch 1 ($M_S$) and epoch 5 ($M_E$). These checkpoints differ in representation and overall performance. In Fig. 2, we showed the results from applying `NMF` to this setting. Here, we also evaluate `SAE` and `KMeans`.

#### A.1.1   `SAE` and `KMeans` Fail to Explain Representational Differences

In Fig. A1 we visualize the explanations generated by `SAE` and `KMeans`. We find that both methods fail to generate explanations that can help us understand the difference between the two representations. The `SAE` generates confusing explanations that may even be misleading. Surprisingly, the `SAE` explanations for $M_S$ are less mixed than $M_E$, suggesting $M_S$ has a better separated representational space, which we know to be incorrect. This is likely a result of random variations in the concepts discovered by the SAE, a phenomena also observed in [14]. `KMeans`, like `NMF`, generates indistinguishable explanations for both representations. This is due to the images near the centroids of similar representations being effectively the same, since these are regions in which model confidence is higher.

#### A.1.2   General Issues with Interpreting Dictionary Learning Methods.

There are two critical issues with interpreting explanations from dictionary learning methods. We visualize these issues in Fig. A2. We show that an explanation consisting of images corresponding to the maximum coefficients of a concept is an *incomplete* explanation. This is because concepts can be *polysemantic* and encode multiple visual features. Polysemantic concepts will have relatively large coefficients for multiple groups of semantically related images. However, by only visualizing the top-k images for a concept vector, we miss important information about other images that shares the same concept vector. These "other" images, with smaller concept coefficients, are critical for understanding the task of comparing two representations, as they may underlie the subtle variations that distinguish one model from another (Fig. A2 A).

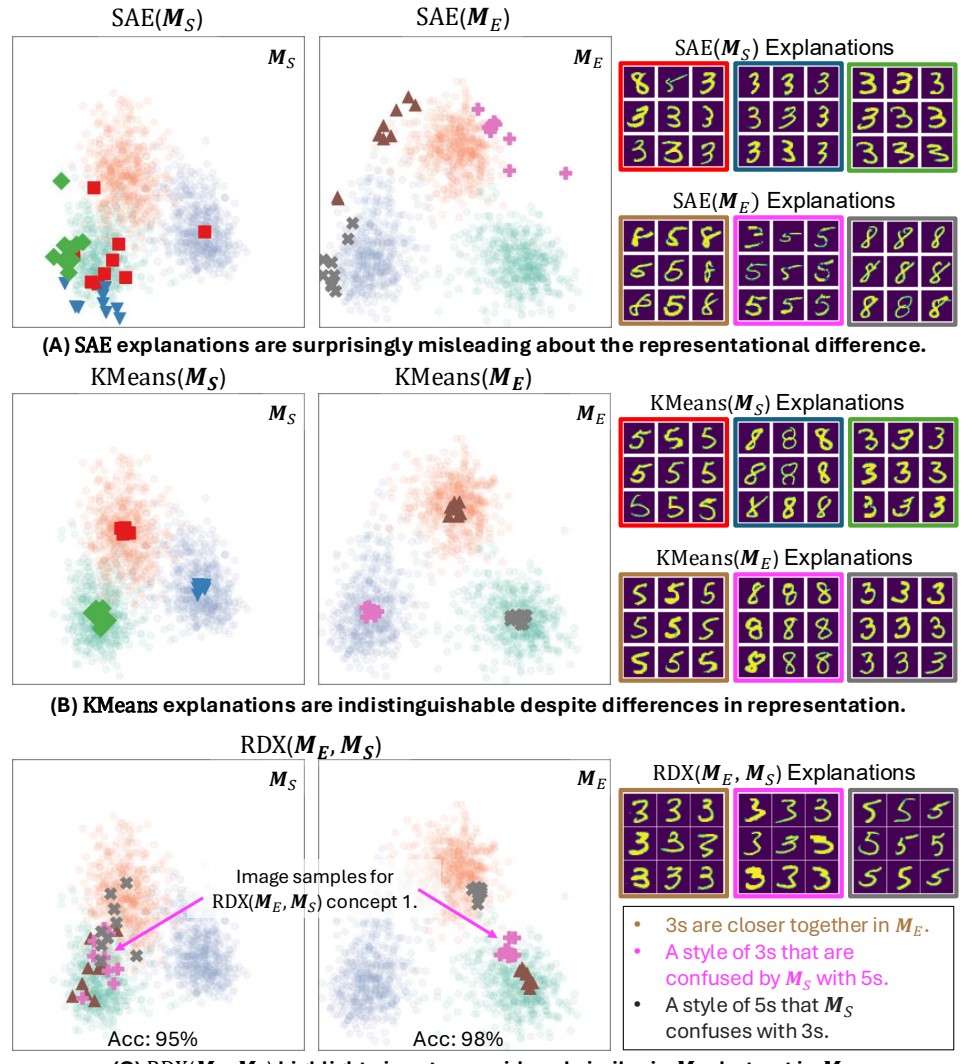

**(A)** SAE **explanations are surprisingly misleading about the representational difference.**

**(B)** KMeans **explanations are indistinguishable despite differences in representation.**

**(C)** RDX$(M_E, M_S)$ **highlights inputs considered similar in $M_E$, but not in $M_S$.**

Figure A1: **Comparing** RDX **to** SAE **and** KMeans**.** In Fig. 2, we visualized NMF explanations for two model representations, from a 'strong' $M_S$ and an 'expert' model $M_E$, trained on MNIST-[3,5,8]. Here we show explanations generated using SAE with maximum sampling and KMeans with centroid sampling. **(A)** SAE explanations are confusing and potentially misleading. SAE$(M_S)$ shows mostly 3s in all explanations, whereas SAE$(M_E)$ shows one explanation with mixed 5s and 8s, and two explanations with 5s and 8s respectively. These explanations do not convey which of the two representations is weaker and may even suggest that the $M_S$ is the expert representation. **(B)** KMeans explanations are indistinguishable. Given that these two representations are highly similar, the centroids for the clusters in both representations are nearly the same. **(C)** RDX$(M_E, M_S)$ shows explanations that helps us understand that $M_E$ does a better job of grouping 3s and 5s than $M_S$, matching the known difference between the two models. The lack of an explanation for 8s suggest that $M_S$ is relatively better at distinguishing 8s. We confirm this in Appendix A.1.4.

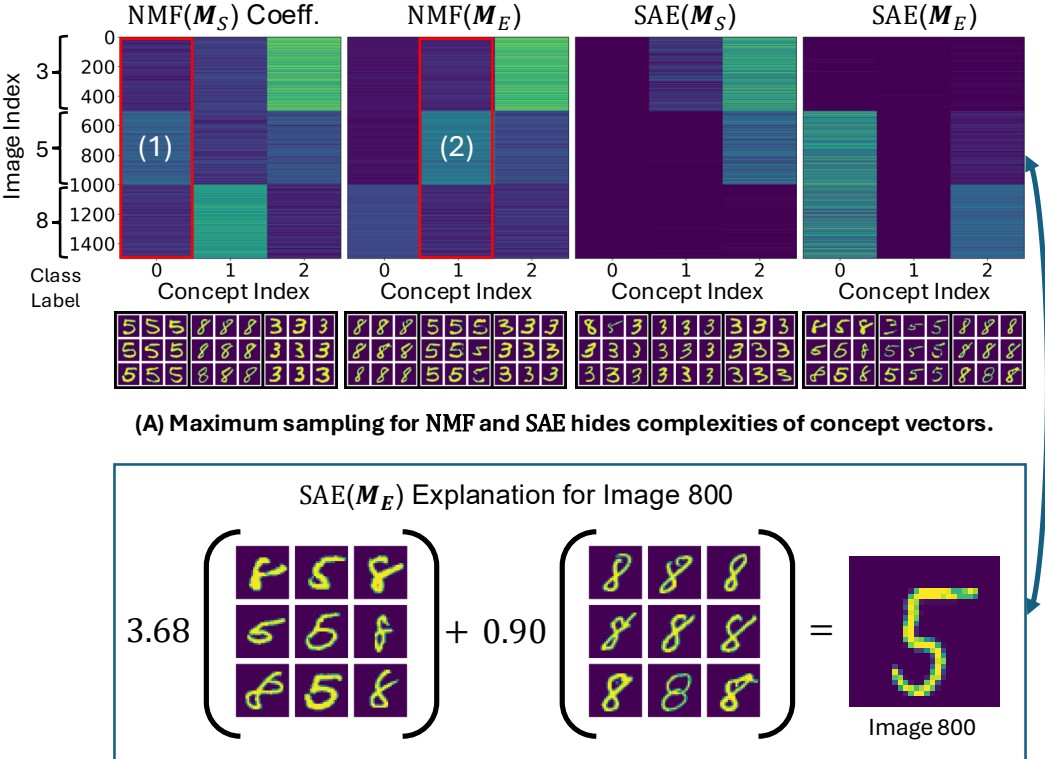

**(A) Maximum sampling for NMF and SAE hides complexities of concept vectors.**

**(B) Weighted combinations of maximally sampled explanations are unintuitive.**

Figure A2: **Interpreting Dictionary Learning Concepts.** We identify two issues with dictionary learning methods for XAI that make them challenging to understand. These results are from the same experiment as in Sec. 4.2 and Appendix A.1. Maximum sampling to explain concept vectors hides important nuances in model behavior. In (1) and (2), highlighted in red, we can see that $M_S$ and $M_E$ both encode roughly the same concept, with maximum activations for images of 5s (indices 500-1000) and weaker activations for images of 3s and 8s (indices 0-500 and 1000-1500). The generated explanations for both concepts show 5s. However, we can see that the activations for 3s and 8s are relatively lower in (2) than the activations for 3s and 8s in (1). These subtle nuances are critical for understanding how the two models behave differently, but are completely lost in the existing approaches for generating explanations. Thus, existing explanations for dictionary learning concepts are *incomplete.* **(B)** Analyzing a single image through the lens of these concepts is extremely challenging. Users are tasked with using incomplete explanations of a concept in a weighted sum with un-intuitive coefficients. For example, image 800 is an image of a "standard" 5, but is encoded by a weighted combination over a concept of "strange" 5s and 8s and normal 8s. See Appendix A.1.2 for a discussion on the impact of monosemanticity and polysemanticity in this context.

### A.1.3 Discussion of the Feasibility and Sufficiency of Monosemanticity

Importantly, the issues with polysemanticity raises questions about the feasibility and sufficiency of decomposing models into monosemantic concepts. Monosemantic concepts are defined as concepts that have a single, unambiguous meaning and extracting them are the goal of sparse autoencoder methods for XAI [9, 17, 52]. Consider a concept vector that encodes the concept of "roundness". This concept is neither monosemantic nor polysemantic, as it is too ambiguous for monosemanticity, but not disparate enough to be polysemantic. On a dataset of objects that are interpolations from squares to circles, this concept would react to all round objects, but most strongly to circles. A maximally sampled explanation for this concept could easily mislead the user into believing that the concept reacts *only* for circles. Trying to convert the "roundness" concept into several discrete monosemantic concepts that only react to well-defined shapes leads to questions about the boundaries of the discretization and the number of concepts that can be meaningfully analyzed by a human. When comparing two models that share the "roundness" concept, differences in discretization could lead to partially overlapping concepts, such as those seen in [28].

When comparing two representations it is sometimes necessary to analyze specific images that the two representations disagree upon. When applying dictionary-learning based methods to understand what concepts make up an image, users are required to mentally perform a weighted combination over incomplete concept explanations (Fig. A2 B). This task is un-intuitive and imprecise in the context of a single model. If the concepts for the two models being compared are even slightly different, this task becomes essentially impossible. Notably, this issue persists even if concepts are monosemantic since most images are likely to contain several concepts.

### A.1.4 Using RDX to Discover Concepts Specific to $M_E$

RDX is not a symmetric method. In Fig. A1 C we visualize the second direction $\text{RDX}(M_E, M_S)$. These explanations reveal images considered similar in $M_E$, but not in $M_S$. These explanations show that the expert model $M_E$ is able to group challenging images of the same digit that $M_S$ is unable to. Additionally, we note that the explanations exclude 8, suggesting that the difference in similarities between images of 8 in $M_E$ and $M_S$ is smaller than the difference in similarities for images of 3 and 5. Indeed, the prediction agreement for linear classifiers trained on these two representations is 95% on 3, 95% on 5, and 98% on 8, confirming our insight from the RDX explanations.

## A.2 Additional Results for Recovering "Known" Differences

We describe the modifications to models in "known" difference comparisons in Table A8. Comparison details are found in Table A11.

### MNIST

In Fig. A3, we visualize the explanations from RDX, KMeans, and NMF for $M_{49}$ vs. $M_b$, $M_{35}$ vs. $M_{49}$, and $M_{\leftrightarrow}$ vs. $M_{\rightleftarrows}$.

In all comparisons, RDX explanations clearly show the expected difference between the two representations. In contrast, KMeans and NMF generate unfocused explanations that are often indistinguishable from each other. At best, we find that the baseline approaches may contain 2/6 explanations focused on the known difference between models.

### CUB PCBM

In Fig. A4 and Fig. A5 we visualize five explanations for comparing $C_{A-S}$ vs. $C_A$ and $C_{A-YB}$ vs. $C_A$ using RDX and a baseline method. $C_A$ is the CUB PCBM concept vector with all concepts retained. $C_{A-S}$ removes the spotted wing concept from the concept vector and $C_{A-YB}$ removes the yellow back concept from the concept vector. We expect that explanations are composed of images that contain these concepts. In both comparisons, the baseline method (CNMF or SAE) generates indistinguishable and unfocused explanations that provide no insight about the known differences. This limitation arises because the representational change induced by removing a single component from the concept vector is extremely subtle. As a result, CNMF and SAE primarily capture the dominant conceptual directions within the representation spaces of both models and remain largely insensitive to these smaller, yet important, representational differences. On the other hand, RDX explanations focus on differences and can reveal interesting insights about the impact of removing a concept. In Fig. A5, the RDX explanations help teach us about how the PCBM uses the "yellow-back" concept. On first glance, the model without the yellow-back concept ($C_{A-YB}$) appears to do a better job of grouping colorful yellow/red birds. When inspected more closely, it becomes clear that there is a mixture of birds with black faces and colored backs and birds with red/yellow faces and black backs. This indicates that the yellow-back concept is used as a fine-grained discriminator between these two color patterns. It also indicates that the PCBM model may be suffering from leakage [19] and does not discriminate between bright red and yellow colors. In the other direction, we see that the yellow-back concept helps organize birds with colorful backs into organized groups (explanations 3-5), but also helps organize "regular" birds into well-separated groups (explanations 1-2).

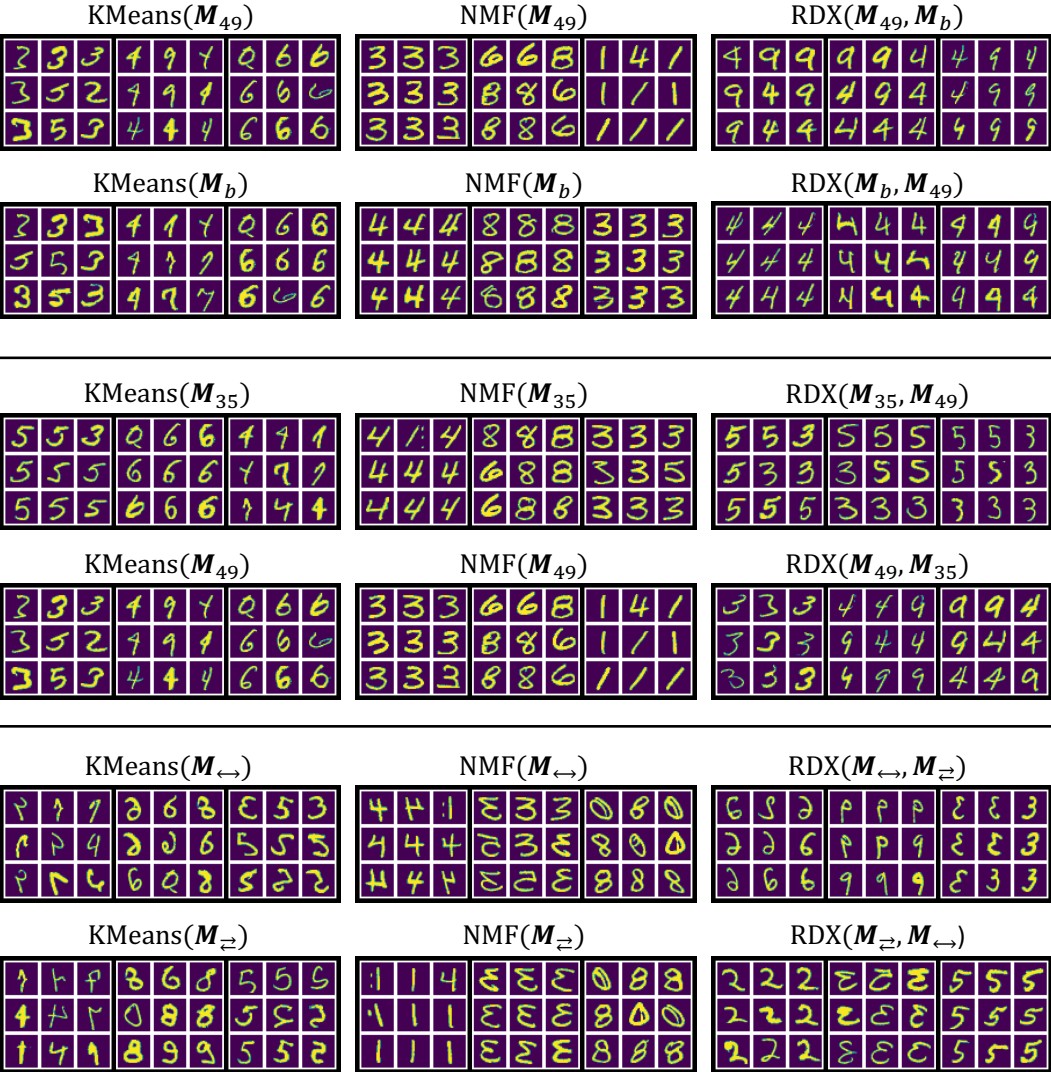

Figure A3: **Additional explanations for MNIST comparisons.** Modifications are described in detail in Table A8. **(Top)** We compare $M_{49}$ (mixes 4s and 9s) to $M_b$ (no modifications). We observe that RDX generates explanations focused on the known difference. KMeans and NMF have 1/6 explanations related the known difference. **(Middle)** We compare $M_{35}$ (mixes 3s and 5s) to $M_{49}$ (mixes 4s and 9s). RDX conveys the modifications made to both models, specifically $M_{35}$ mixes 3s and 5s and that $M_{49}$ mixes 4s and 9s. Also, it shows that $M_{49}$ organizes 3s better than $M_{35}$. KMeans has one explanation related to the known differences, while NMF has none. **(Bottom)** We compare $M_{\leftrightarrow}$ (mixed flipped and unflipped digits) to $M_{\rightleftarrows}$ (separated flipped and unflipped digits). RDX reveals mixing between flipped and unflipped 6s, 9s and 3s in $M_{\leftrightarrow}$ and no mixing for 2s, flipped 3s and 5s in $M_{\rightleftarrows}$. KMeans explanations are confusing. NMF has 2/6 explanations that align with the known difference. See Appendix A.2 for discussion.

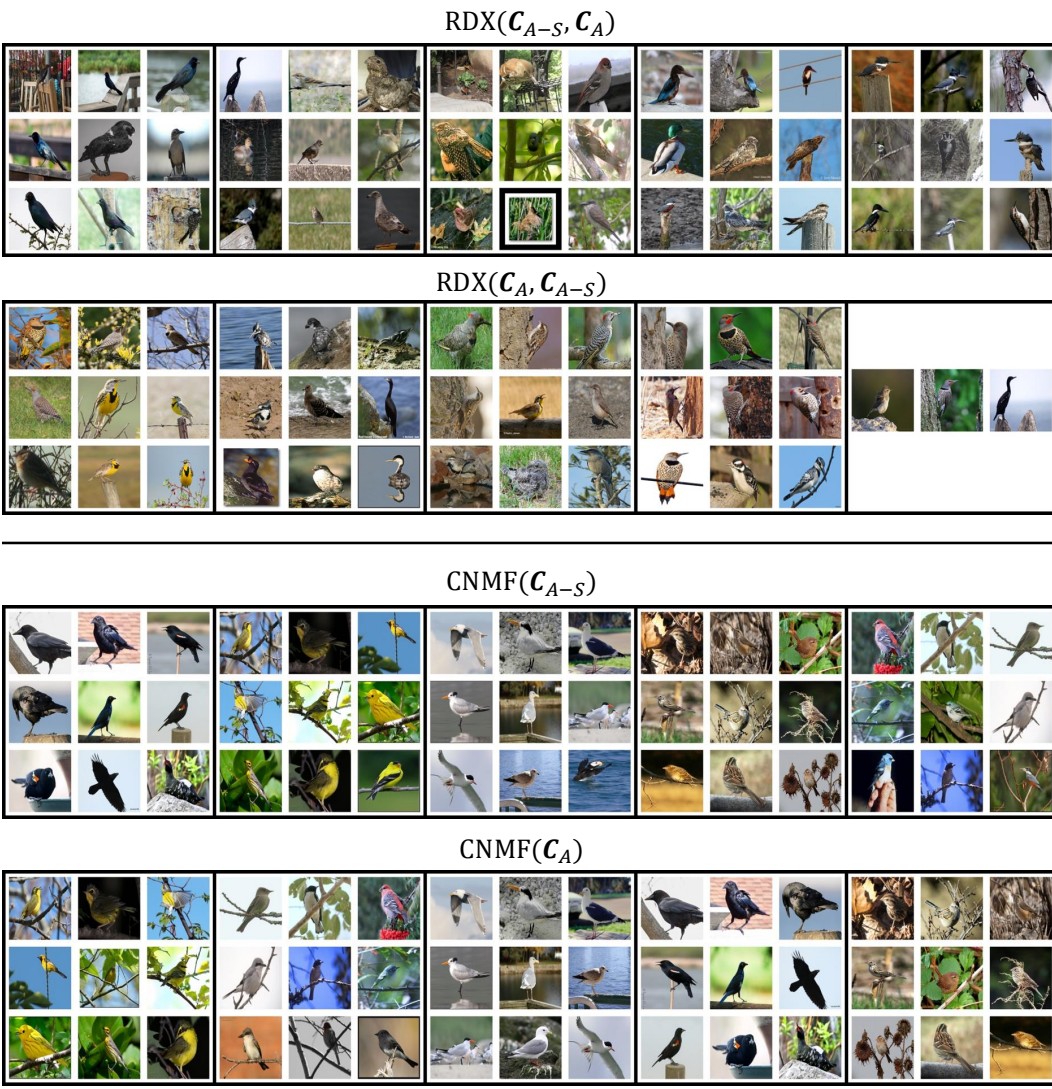

Figure A4: **Explanations for the "Spotted Wing" concept.** We selected a few explanation grids to show in Fig. 5. Here we visualize all five explanations generated by two XAI methods, best viewed zoomed in. **(Top)** In row 1, the RDX explanation grids show birds with and without spotted wings mixed together. In row 2, the explanation grids are predominantly made up of birds with spotted wings. In explanation five, we see that RDX can generate clusters with too few images to generate a full grid. **(Bottom)** In both rows, each CNMF explanation grids shows a different kind of bird, unrelated to the known difference. For example, we can see concepts for yellow birds, seabirds, and black birds. The explanations for both representations are indistinguishable. See Appendix A.2 for discussion.

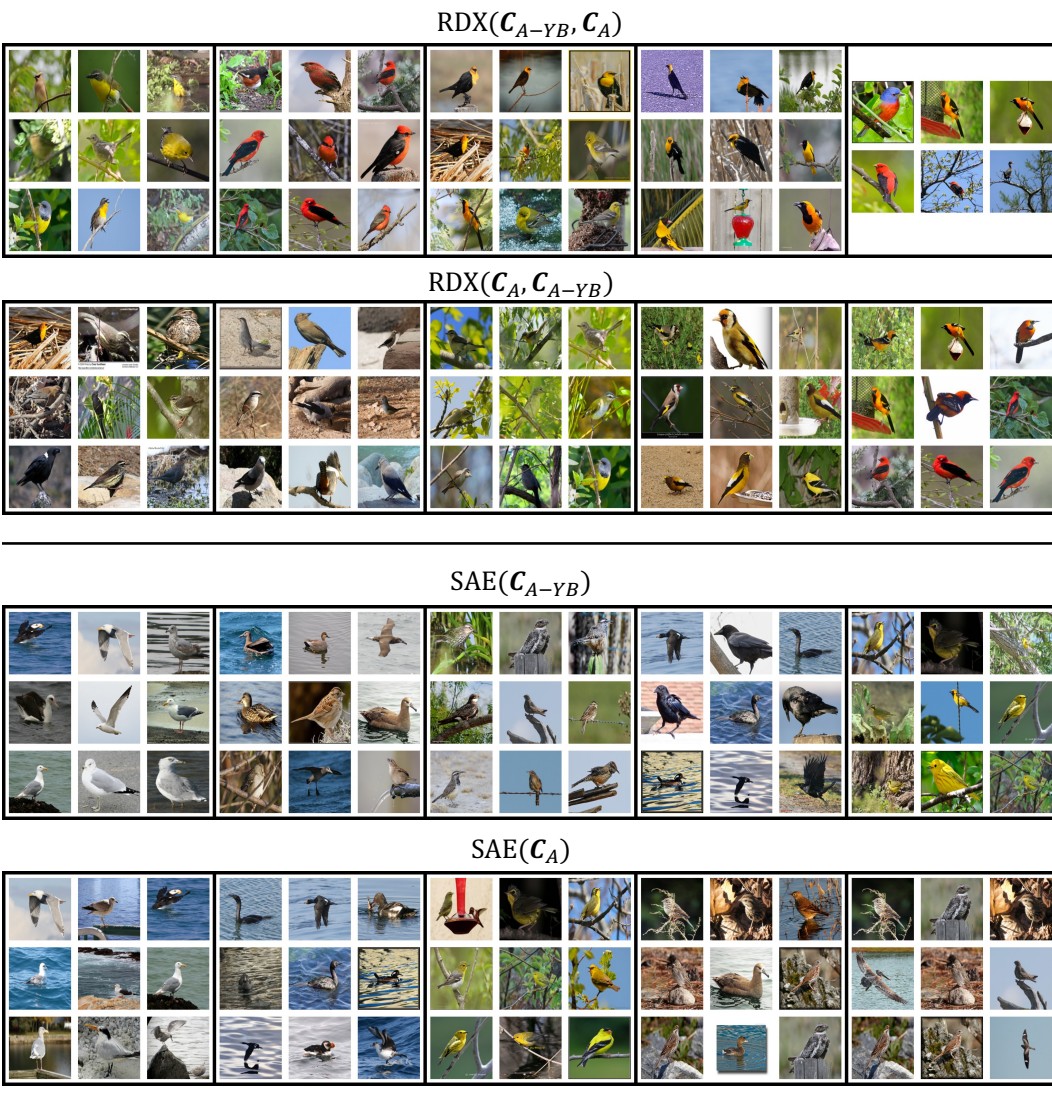

Figure A5: **Explanation for the "Yellow Back" concept.** We visualize explanations from comparing $C_{A-YB}$ vs. $C_A$. **(Top)** In row 1, the RDX explanation grids reveal a mixture of red and yellow birds. Closer inspection shows that these grids, especially explanation 2 and 4, contain birds with yellow or red backs intermixed with those having black backs. For instance, in explanation 4, the central bird has a black back, and the lower-left bird has a yellow back. In contrast, row 2 shows that $C_A$ does not mix birds *with* and *without* colored backs within the same concept. In explanations 1–2, $C_A$ isolates birds with black or gray backs into distinct concepts, while in explanations 1–3, it groups multi-colored birds with tan, yellow, and red backs together. These patterns suggest that the "Yellow Back" concept may serve to disambiguate birds with similar color distributions but differing spatial arrangements of those colors. **(Bottom)** In both rows, each SAE explanation grids shows concepts that correspond to different bird types, unrelated to the known difference. The explanations for both representations are also indistinguishable. See Appendix A.2 for discussion.

## A.3 Additional Results for Discovering "Unknown" Differences

Table A1: **Linear probe accuracy across datasets described in Table A11.**

|  | Mittens | Primates | Maples | Corvids |
|---|---|---|---|---|
| **DINO** ($M_D$) | 0.933 | 0.918 | – | – |
| **DINOv2** ($M_{D2}$) | **0.965** | **0.921** | – | – |
| **CLIP** ($M_C$) | – | – | 0.752 | **0.790** |
| **CLIP-iNat** ($M_{CN}$) | – | – | **0.796** | 0.787 |

We visualize complete `RDX` results for four comparisons using the representational alignment step from Sec. 3.4:

1. $M_D$ vs. $M_{D2}$ on Mittens (Fig. A7)

2. $M_D$ vs. $M_{D2}$ on Primates (Fig. A8)

3. $M_C$ vs. $M_{CN}$ on Maples (Fig. A9)

4. $M_C$ vs. $M_{CN}$ on Corvids (Fig. A10)

See Table A11, for a description of the datasets and Table A9 for details about the models. Although explaining classifier predictions is not the goal of `RDX`, we train a linear classifier on these representations to gain an insight into the quality of their organization and to assist in interpretation (Table A1). Training details are provided in Appendix C.1. We use the classifier accuracies and predictions as supplemental information to understand representational differences.

In all representational difference comparisons, `RDX` reveals interesting insights about differences in model representations.

In the first two comparisons (Mittens and Primates), classification performance indicates that the representations are well-organized and we are able to easily interpret the discovered concepts without using the dataset labels.

In the next two comparisons (Maples and Corvids), we evaluate how `RDX` performs when classifier performance is much lower. We expect these representations to be more poorly organized and subsequently more challenging to interpret. In the first comparison (Maples), we select a comparison where there is a large performance difference (4%). In the second (Corvids), we explore a setting in which fine-tuning CLIP on iNaturalist images did not improve the quality of the representation, although it may have changed it.

### Mittens

We find that DINOv2 does a better job of organizing mittens by their orientation (see Fig. A7). It also has a unique concept for children's mittens that is not present in DINO. On the other hand, DINO has unique concepts for children around Christmas related items and Christmas items on their own. These two concepts appear to be entangled in DINOVv2.

### Primates

We notice that DINO contains four unique concepts that appear to fixate on secondary characteristics (see Fig. A8). Explanation 1 seems to react to siamangs on grass, explanation 3 picks up on lower quality images, like screenshots from a video or images taken through enclosure glass, explanation 4 contains scenes of branches and greenery where primates are distant, and explanation 5 contains scenes of a variety of primates behind fencing. In contrast, DINOv2 explanations tend to be focused on the primate type. For example, explanation 1 also picks up on siamangs, but in a variety of environments, suggesting that DINOv2 may be less biased by background. Explanation 3 shows gibbons swinging in trees and explanation 5 shows spider monkeys in trees, these two concepts are entangled in DINO, suggesting DINO is more sensitive to the background/activity of the monkeys than the species.

Table A2: Ground truth label counts for explanations on Maples.

| Maple Type | $\text{RDX}(\boldsymbol{M}_C, \boldsymbol{M}'_{CN})$ | | | | | $\text{RDX}(\boldsymbol{M}_{CN}, \boldsymbol{M}'_C)$ | | | | |
| --- | --- | --- | --- | --- | --- | --- | --- | --- | --- | --- |
| | E1 | E2 | E3 | E4 | E5 | E1 | E2 | E3 | E4 | E5 |
| Norway Maple | 1 | 1 | 2 | 0 | 0 | 9 | 0 | 1 | 0 | 0 |
| Silver Maple | 3 | 3 | 1 | 2 | 1 | 0 | 0 | 7 | 0 | 9 |
| Sugar Maple | 3 | 3 | 4 | 0 | 0 | 0 | 9 | 0 | 0 | 0 |
| Red Maple | 2 | 2 | 2 | 7 | 8 | 0 | 0 | 1 | 9 | 0 |

**Maples**

Fine-grained maple tree classification is a challenging task which is beyond the skill of most people. The clear performance gap between model's indicates that CLIP-iNat has learned important features, we explore if RDX is able to help us generate hypotheses on what those might be (see Fig. A9). In Fig. 6 we analyzed explanations 4 and 5 from $\text{RDX}(\boldsymbol{M}_{CN}, \boldsymbol{M}'_C)$. This allowed us to hypothesize that CLIP was biased towards encoding Maple leaves by color/season rather than species. Despite this bias, we find that classifiers trained on both representations perform reasonably well on these two image grids. This suggests that this representational difference may not be important for classification. However, we notice that there are differences in the classifier predictions for explanations 2 and 3. We use this information to propose some hypotheses. The dataset labels for explanation 2 indicate that all of the images are sugar maples. CLIP-iNat gets 7/9 correct, while CLIP gets 5/9. With this information, we hypothesize that CLIP-iNat is able to detect sugar maples leaves in images with a variety of seasons, backgrounds, and lighting. In explanation 3, we see young, bushy maple trees around rocks and waterways. The classification labels indicate the majority of these images contain silver maples. CLIP-iNat gets 8/9 correct, while CLIP gets only 4/9 correct. This suggests that CLIP-iNat has learned to associate this visual concept with silver maples and that this is an effective strategy for classification on this dataset. We also observe some higher-level characteristics of the explanations when analyzed with their ground truth labels. Explanations for CLIP-iNat tend to have labels that correspond with one of the ground truth labels, while CLIP does not (Table A2). This make sense, as CLIP-iNat was fine-tuned on a classification dataset.

**Corvids**

Neither representation supports good classification on the challenging Corvids dataset (see Fig. A10). However, RDX reveals some interesting concepts unique to each model. For example, $\text{RDX}(\boldsymbol{M}_C, \boldsymbol{M}'_{CN})$ explanation 4, shows a CLIP specific concept for Corvid footprints. In explanation 5, we see a concept for flocks of crows. In the other direction, CLIP-iNat has a learned a concept for large ravens in natural settings like hillsides or beaches (explanation 1). It has also learned a concept for crows in urban settings like schools, fields and pavement (explanation 2). Additionally, CLIP-iNat makes a stronger distinction between perched crows in urban settings (explanation 4) and flying crows (explanation 5).

**Effects of Alignment**

In Fig. A6, we visualize the effect of alignment when comparing CLIP to CLIP-iNat on the Maples dataset. We see that alignment can result in a significant change in the spectral clusters detected from the affinity matrix. In particular, one discovered concept is only present in the unaligned comparison. This indicates that both representations contain the information to represent the concept, but their initial configurations differ. After alignment, the concepts discovered are more likely to be fundamental differences between the two representations. Although there are some limitations to this interpretation (see Sec. 5), we focus on aligned comparisons in our qualitative plots, as it makes interpretation simpler.

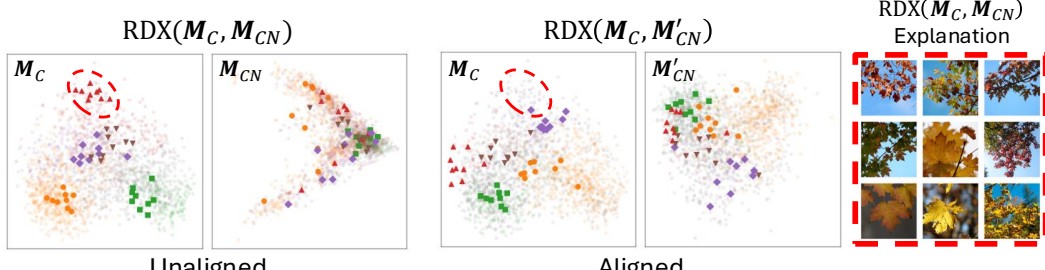

Figure A6: **Effect of Alignment (Appendix A.3). (Top)** We compare CLIP and CLIP-iNat with and without aligning CLIP-iNat (denoted by $M'$ notation). We visualize PCA plots of the representations with cluster membership (light colors) and samples (dark colors). In each direction, the left plot contains the concept "source" representation while the right plot has the selected clusters overlayed on its representation. **(Unaligned, Left)** We generate five spectral clusters, we can see that they group nicely in the left plot and are spread apart in the right plot. We highlight the region of the red cluster for comparison after alignment. **(Aligned, Right)** After alignment, we can see that it is more difficult to get clusters that have large differences in their distances in the two representations. We find that the region that the red cluster came from in the unaligned comparison is no longer selected after alignment. **(Explanation)** The red cluster (unaligned) contains maples in fall foliage. Both networks can represent this concept, although the unaligned CLIP-iNat representation does not prioritize it.

**ChatGPT-4o Analysis**

Analyzing and annotating several explanations for each model is time consuming and cognitively demanding. We explore if ChatGPT-4o [7] is capable of annotating the images for us in the Maples and Corvids comparisons. We use the prompt:

> **Prompt**
>
> I am going to ask you to analyze image grids. You will receive a strip of five image grids. The images in the grid will be from the categories: <category list>. Your task is to concisely describe the consistent features that appear in each image grid. You do not need to use complete sentences. The format of your output should be a dictionary like this. E1: desc1, E2: desc2, E3: desc3, E4: desc4, E5: desc5.

The outputs are provided in the figure captions of Fig. A9 and Fig. A10. We find that the annotations are clear, reasonable and helpful.

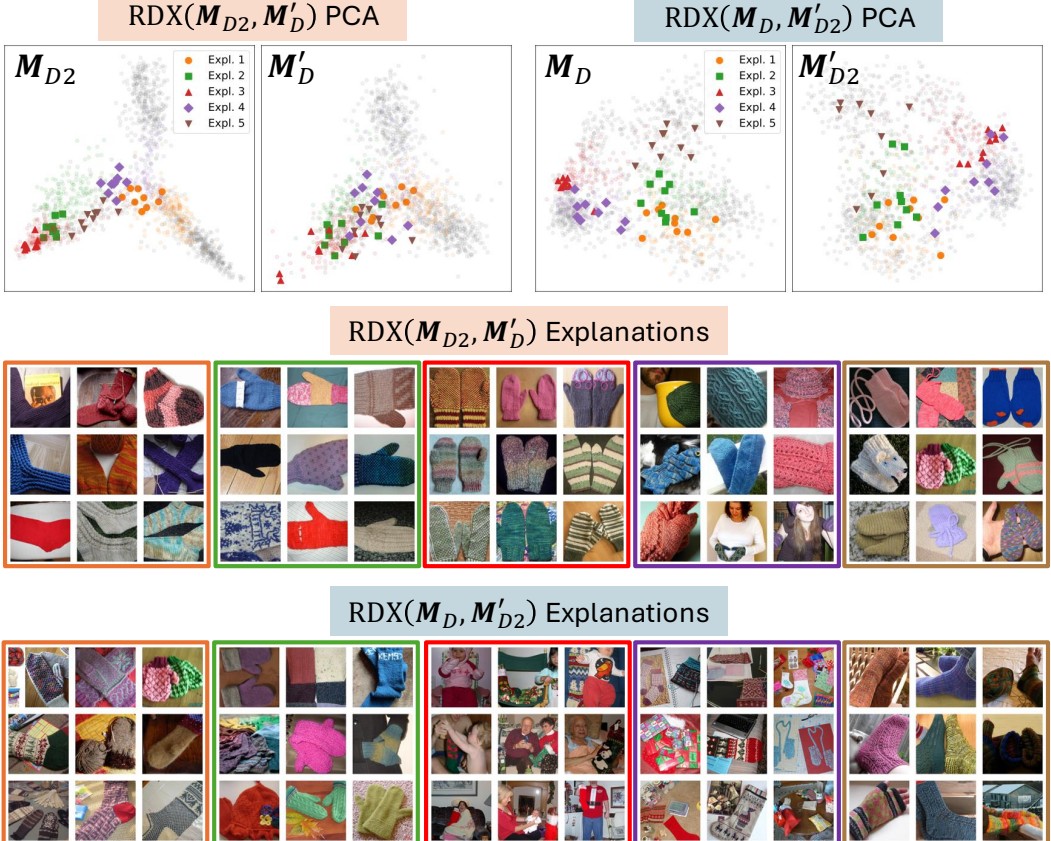

Figure A7: **Investigating DINOv2 vs. DINO on Mittens (aligned).** We visualize RDX difference explanations in both directions on the Mittens dataset (Table A11). This dataset contains images of mittens, socks and Christmas stocking from ImageNet [10]. For example, RDX($\boldsymbol{M}_{D2}, \boldsymbol{M}'_D$) generates explanations for concepts that are in $\boldsymbol{M}_{D2}$ (DINOv2), but not $\boldsymbol{M}'_D$ (aligned DINO). We refer to the explanations as E1 to E5 (left to right). **(Top)** PCA plots of the representations with cluster membership (light colors) and samples (dark colors). In each direction, the left plot contains the concept "source" representation while the right plot has the selected clusters overlaid on its representation. Clusters on the left plot generally appear better organized than in the right plot. (RDX($M_{D2}, M'_D$)) E1: crocheted socks, E2: horizontal mittens, E3: vertical pairs of mittens, E4: crocheted mittens, E5: children's mittens. (RDX($M_D, M'_{D2}$)) E1: multi-colored wool socks, E2: assorted pairs of mittens, E3: children with Christmas decorations, E4: Christmas paraphernalia mittens, and E5: woolen clothes being worn. See Appendix A.3 for interpretation.

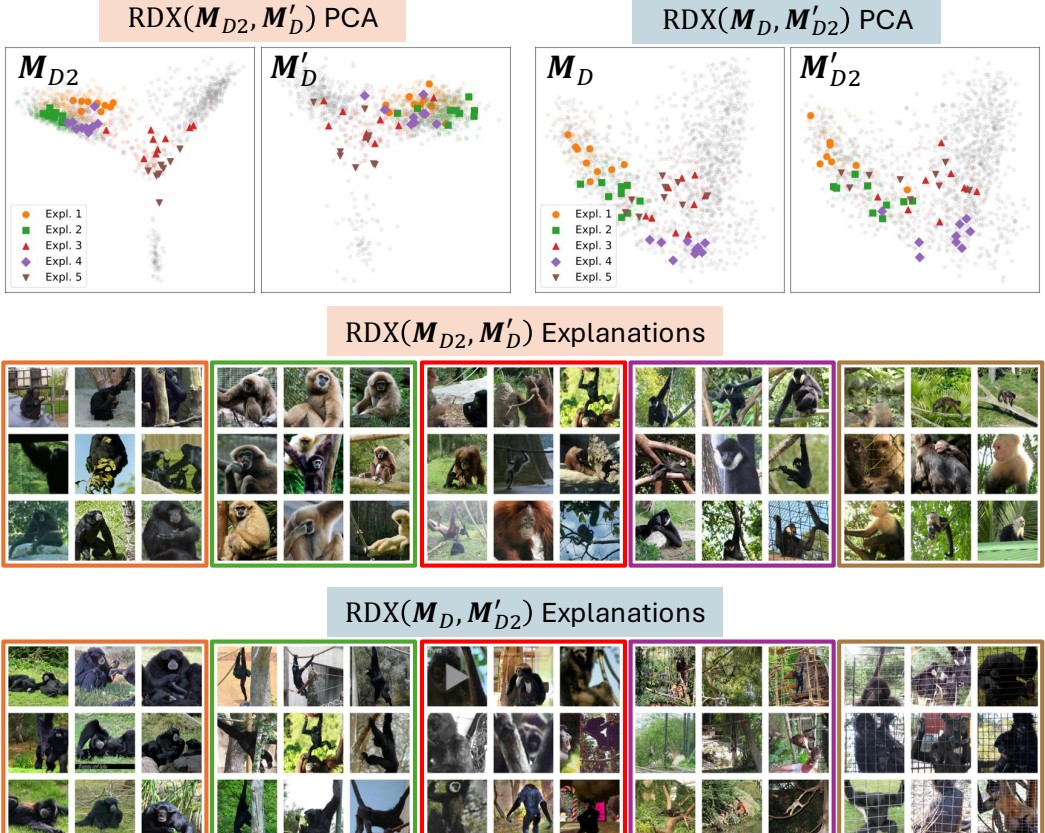

Figure A8: **Investigating DINOv2 vs. DINO on Primates (aligned).** We visualize RDX difference explanations in both directions on the Primates dataset (Table A11). This dataset contains images of gibbons, siamangs and spider monkeys from ImageNet [10]. For example, RDX($M_{D2}, M'_D$) generates explanations for concepts that are in $M_{D2}$ (DINOv2), but not $M'_D$ (aligned DINO). We refer to the explanations as E1 to E5 (left to right). **(Top)** PCA plots of the representations with cluster membership (light colors) and samples (dark colors). In each direction, the left plot contains the concept "source" representation while the right plot has the selected clusters overlayed on its representation. Clusters on the left plot generally appear better organized than in the right plot. (RDX($M_{D2}, M'_D$)) E1: black siamangs in diverse environments, E2: tan gibbons, E3: orangutans and gibbons playing, E4: white-cheeked gibbons, and E5: spider monkeys. (RDX($M_D, M'_{D2}$)) E1: siamangs laying in grass, E2: gibbons swinging, E3: lower resolution images of primates, viewed through glass or from videos, E4: tree environment with distant primates, and E5: assorted primates behind fencing. See Appendix A.3 for interpretation.

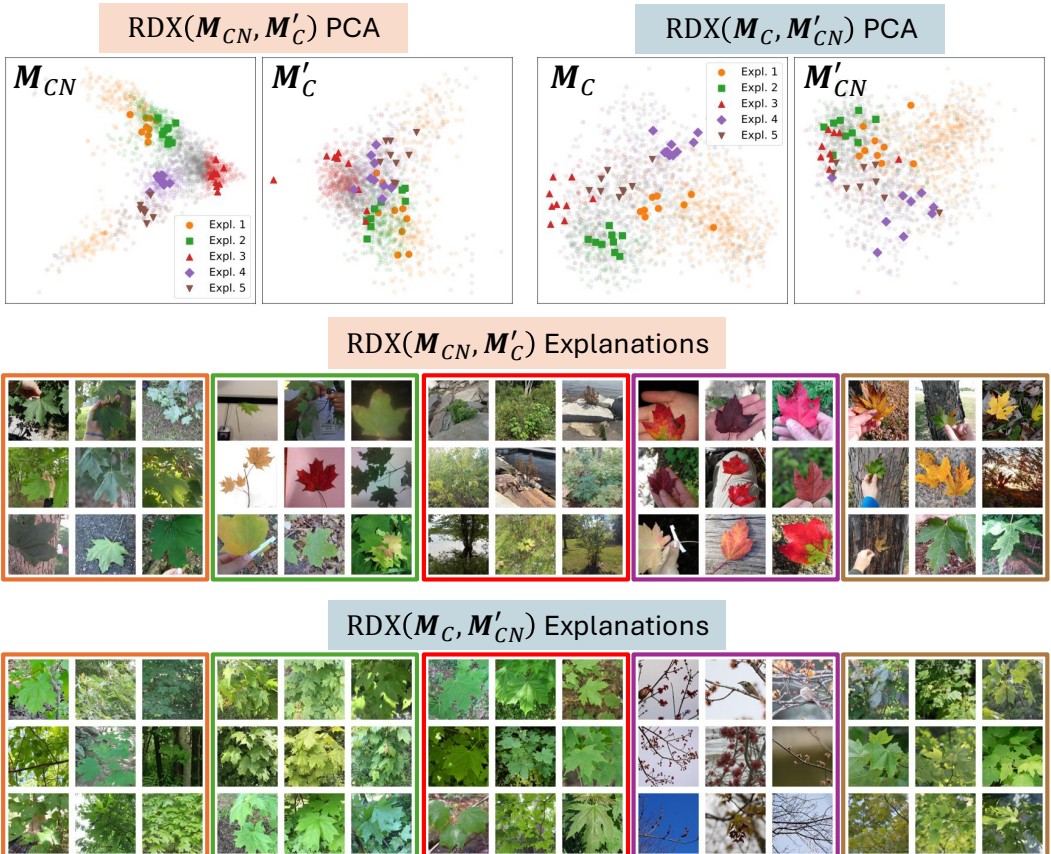

Figure A9: **Investigating CLIP-iNat vs. CLIP on Maples (aligned).** We visualize RDX difference explanations in both directions on the Maples dataset (Table A11). This dataset contains images of red maples, sugar maples, Norway maples, and silver maples from iNaturalist [24]. $\text{RDX}(M_{CN}, M'_{CN})$ generates explanations for concepts that are in $M_{CN}$ (CLIP-iNat), but not $M'_C$ (aligned CLIP). We refer to the explanations as E1 to E5 (left to right). **(Top)** PCA plots of the representations with cluster membership (light colors) and samples (dark colors). In each direction, the left plot contains the concept "source" representation while the right plot has the selected clusters overlaid on its representation. Clusters on the left plot generally appear better organized than in the right plot.

These types of maples have subtle differences beyond the expertise of most people so we use ChatGPT-4o to generate descriptions. ($\text{RDX}(M_{CN}, M'_C)$) E1: "Large, dark green, sharply lobed leaves; smooth surface; some handheld, often against tree bark or forest background", E2: "Varied color (green, red, yellow), symmetric lobes with central point, often single leaves photographed on flat surfaces", E3: "Small clusters of light green to reddish leaves, forest floor or rocky environment, less prominent lobes", E4: "Bright red leaves, often handheld, five lobes with narrow points, smooth margins, clear vein structure", and E5: "Yellow mottled leaves, some black spotting, thick lobes, visible veins, photos taken in autumn light or against tree bark". ($\text{RDX}(M_C, M'_{CN})$) E1: "Leaves with deep sinuses, bright green, flat edges, consistent lighting, often low to ground or with visible bark", E2: "Yellow-green foliage, broad flat leaves with few teeth, tree clusters with hanging leaves, slight curl", E3: "Five-lobed leaves, medium green, fine-toothed edges, spread flat, some variation in lighting and angle", E4: "Red spring buds and samaras, no full leaves visible, bare branches, sky background, some birds", and E5: "Light green leaves with coarsely toothed edges, translucent lighting, some purplish tinge in parts, lobed leaves". See Appendix A.3 for interpretation.

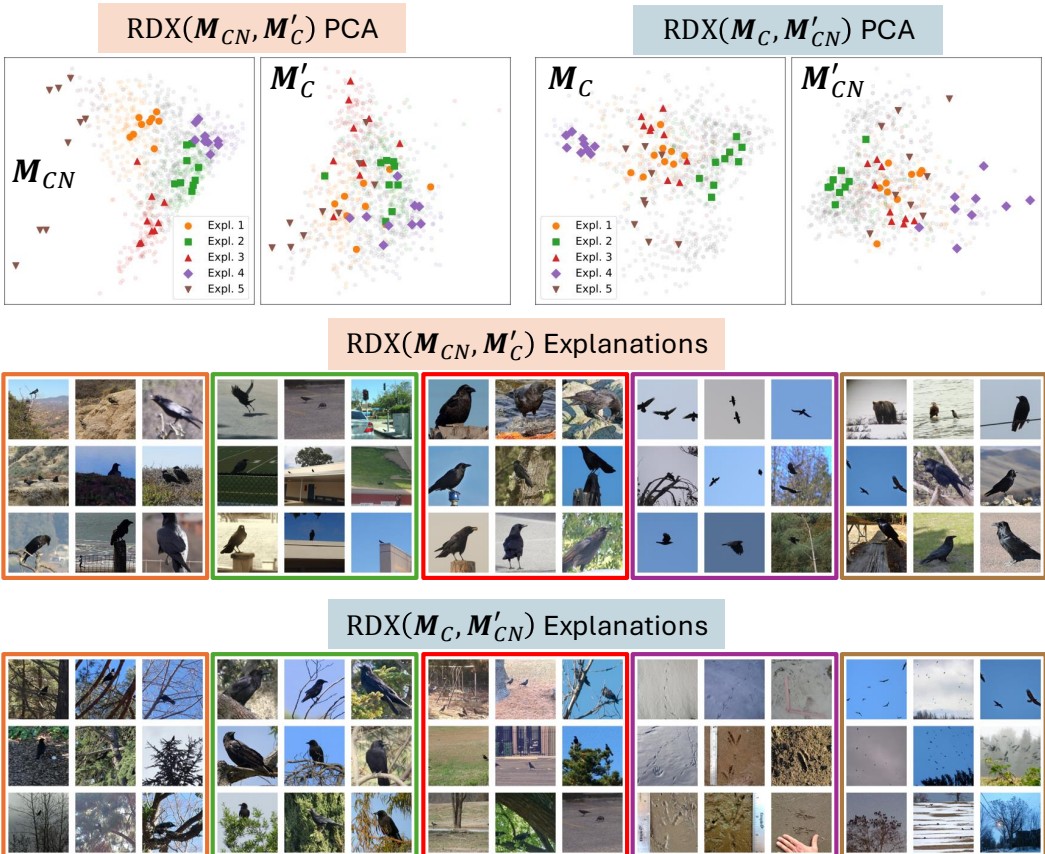

Figure A10: **Investigating CLIP-iNat vs. CLIP on Corvids (aligned).** We visualize RDX difference explanations in both directions on the Corvids dataset (Table A11). This dataset contains images of crows and ravens from iNaturalist [24]. For example, RDX($M_{CN}, M_{CN}'$) generates explanations for concepts that are in $M_{CN}$ (CLIP-iNat), but not $M_C'$ (aligned CLIP). We refer to the explanations as E1 to E5 (left to right). **(Top)** PCA plots of the representations with cluster membership (light colors) and samples (dark colors). In each direction, the left plot contains the concept "source" representation while the right plot has the selected clusters overlaid on its representation. Clusters on the left plot generally appear better organized than in the right plot.

These types of corvids have subtle differences beyond the expertise of most people so we use ChatGPT-4o to generate descriptions. (RDX($M_{CN}, M_C'$)) E1: 'Birds in arid or rocky environments; perched or flying; often alone or in small groups; slimmer builds; medium size; matte black feathers", E2: 'Urban and suburban settings; birds near buildings, fences, and pavement; typically foraging; in pairs or groups; more compact build", E3: 'Close-up or detailed views of large, shaggy birds; prominent beaks and throat hackles; perched or interacting with environment", and E4: 'Birds flying in sky; high contrast silhouettes; open sky backgrounds; wing shapes and flight patterns emphasized", E5: 'Birds with other wildlife (e.g. bear, eagle); perched alone or with others; prominent size; thick bills and throat feathers". (RDX($M_C, M_{CN}'$)) E1: "Birds in wooded or forested environments; perched on branches; medium size; matte black feathers; mostly solitary or in pairs", E2: 'Birds on open branches or tall perches; slightly larger size; thick beaks; prominent neck feathers (hackles); more upright posture", E3: 'Birds on ground in urban/park environments; sparse trees; usually in small groups; foraging or walking", E4: 'Footprints in mud, sand, or snow; distinct three-toed tracks; measurement tools in several images; variable substrate", and E5: 'Flocks of birds flying or perched in large groups; sky or treetops visible; misty or open-air environments". See Appendix A.3 for interpretation.

## A.4 Additional Metrics and Detailed Results for Comparison Experiments

In addition to our BSR metric, we compute the Redundancy, Clarity, and Polysemanticity metrics from SemanticLens [12]. These metrics use a model, OpenCLIP [23], to evaluate the explanation grids for each concept. Details on how each metric is computed are provided in Appendix B.2. We modify the Redundancy metric from the original work [12], such that it is designed to compare the redundancy of concepts *across* models. Lower redundancy indicates that the concepts discovered for each model are less similar, while higher redundancy indicates that the concepts for each model are more similar. The results are presented in Tables A3, A4, A5 and A6. In Appendix C, we provide more details on the models (Table A8) and datasets (Table A11) used in each comparison. Comparisons are written in a single direction, but results are reported for both directions. The ′ is used to indicate experiments conducted after an alignment step has been performed.

RDX is designed to isolate and visualize concepts unique to each model in a comparison. Thus, we expect and find that it performs the best on the BSR and Redundancy metrics (Tables A3 and A4. However, isolating model differences may also result in highlighting parts of the representation that are more likely to be complex (higher polysemanticity) and less clear (lower clarity). Therefore, it is not surprising that RDX does not have an advantage on either of these metrics (Tables A5 and A6), although it performs comparably to other baseline methods.

Additionally, there are some caveats to interpreting the SemanticLens metrics. These metrics assume that the generalist model (OpenCLIP) is an unbiased evaluator, but it is likely that OpenCLIP has its own biases that will affect each method's scores. This is especially true when generating explanations for complex, niche datasets that are out of domain for OpenCLIP. We can see this bias in the MNIST dataset comparisons under the redundancy metric (Table A4, first five rows). Despite the RDX concepts for each model being semantically different (Fig. A3), OpenCLIP is incapable of distinguishing the nuanced differences that are plainly visible to human observers, resulting in high redundancy scores for all methods. A benefit of our BSR metric (Table A3, first five rows) is that it does not depend on a separate generalist model and better matches the qualitative performance differences observed in Fig. A3.

Table A3: **BSR results across all comparison experiments.** The best (highest) mean value per row is bolded (ties bolded). RDX has the best scores on the BSR metric.

| Comparison | RDX | KMeans | TKSAE | NMF | SAE | NLMCD |
|---|---|---|---|---|---|---|
| $M_{35}$ vs. $M_b$ | $\mathbf{0.83} \pm 0.03$ | $0.62 \pm 0.00$ | $0.61 \pm 0.05$ | $0.58 \pm 0.06$ | $0.57 \pm 0.06$ | $0.52 \pm 0.03$ |
| $M_{49}$ vs. $M_b$ | $\mathbf{0.86} \pm 0.02$ | $0.67 \pm 0.03$ | $0.45 \pm 0.05$ | $0.57 \pm 0.03$ | $0.56 \pm 0.04$ | $0.52 \pm 0.08$ |
| $M_{35}$ vs. $M_{49}$ | $\mathbf{0.87} \pm 0.00$ | $0.68 \pm 0.01$ | $0.65 \pm 0.02$ | $0.50 \pm 0.04$ | $0.56 \pm 0.05$ | $0.56 \pm 0.06$ |
| $M_{\leftrightarrow}$ vs. $M_{\rightleftarrows}$ | $\mathbf{0.90} \pm 0.05$ | $0.79 \pm 0.06$ | $0.69 \pm 0.03$ | $0.59 \pm 0.07$ | $0.51 \pm 0.04$ | $0.57 \pm 0.02$ |
| $M_{\updownarrow}$ vs. $M_{\uparrow\downarrow}$ | $\mathbf{0.91} \pm 0.04$ | $0.72 \pm 0.07$ | $0.56 \pm 0.13$ | $0.72 \pm 0.02$ | $0.69 \pm 0.04$ | $0.54 \pm 0.19$ |
| $C_{A-S}$ vs. $C_A$ | $\mathbf{0.71} \pm 0.12$ | $0.30 \pm 0.00$ | $0.32 \pm 0.14$ | $0.38 \pm 0.11$ | $0.38 \pm 0.19$ | $0.27 \pm 0.05$ |
| $C_{A-YB}$ vs. $C_A$ | $\mathbf{0.66} \pm 0.08$ | $0.21 \pm 0.05$ | $0.33 \pm 0.11$ | $0.38 \pm 0.10$ | $0.37 \pm 0.08$ | $0.25 \pm 0.05$ |
| $C_{A-YC}$ vs. $C_A$ | $\mathbf{0.64} \pm 0.13$ | $0.30 \pm 0.00$ | $0.36 \pm 0.18$ | $0.41 \pm 0.13$ | $0.38 \pm 0.09$ | $0.32 \pm 0.15$ |
| $C_{A-E}$ vs. $C_A$ | $\mathbf{0.69} \pm 0.11$ | $0.21 \pm 0.01$ | $0.31 \pm 0.16$ | $0.38 \pm 0.12$ | $0.32 \pm 0.06$ | $0.39 \pm 0.15$ |
| $C_{A-D}$ vs. $C_A$ | $\mathbf{0.68} \pm 0.12$ | $0.21 \pm 0.07$ | $0.35 \pm 0.06$ | $0.40 \pm 0.01$ | $0.32 \pm 0.12$ | $0.26 \pm 0.11$ |
| $M_D$ vs. $M_{D2}$ ($MT$) | $\mathbf{0.92} \pm 0.01$ | $0.88 \pm 0.01$ | $0.76 \pm 0.02$ | $0.68 \pm 0.05$ | $0.59 \pm 0.04$ | $0.53 \pm 0.11$ |
| $M_D$ vs. $M_{D2}$ ($B$) | $\mathbf{0.88} \pm 0.02$ | $0.87 \pm 0.07$ | $0.66 \pm 0.07$ | $0.78 \pm 0.06$ | $0.56 \pm 0.07$ | $0.50 \pm 0.04$ |
| $M_D$ vs. $M_{D2}$ ($D$) | $\mathbf{0.93} \pm 0.04$ | $0.81 \pm 0.02$ | $0.71 \pm 0.05$ | $0.66 \pm 0.01$ | $0.48 \pm 0.03$ | $0.55 \pm 0.14$ |
| $M_D$ vs. $M_{D2}$ ($P$) | $\mathbf{0.92} \pm 0.03$ | $0.79 \pm 0.02$ | $0.74 \pm 0.03$ | $0.76 \pm 0.05$ | $0.55 \pm 0.01$ | $0.42 \pm 0.14$ |
| $M_C$ vs. $M_{CN}$ ($G$) | $\mathbf{0.97} \pm 0.01$ | $0.93 \pm 0.02$ | $0.80 \pm 0.08$ | $0.82 \pm 0.06$ | $0.71 \pm 0.07$ | $0.51 \pm 0.03$ |
| $M_C$ vs. $M_{CN}$ ($C$) | $\mathbf{0.93} \pm 0.03$ | $0.86 \pm 0.03$ | $0.76 \pm 0.07$ | $0.75 \pm 0.05$ | $0.65 \pm 0.01$ | $0.53 \pm 0.09$ |
| $M_C$ vs. $M_{CN}$ ($MP$) | $\mathbf{0.98} \pm 0.01$ | $0.95 \pm 0.02$ | $0.87 \pm 0.03$ | $0.80 \pm 0.07$ | $0.64 \pm 0.05$ | $0.46 \pm 0.23$ |
| $M_D$ vs. $M'_{D2}$ ($MT$) | $\mathbf{0.90} \pm 0.01$ | $0.56 \pm 0.05$ | $0.46 \pm 0.09$ | $0.49 \pm 0.06$ | $0.45 \pm 0.12$ | $0.52 \pm 0.07$ |
| $M_D$ vs. $M'_{D2}$ ($B$) | $\mathbf{0.92} \pm 0.01$ | $0.58 \pm 0.07$ | $0.46 \pm 0.05$ | $0.47 \pm 0.01$ | $0.40 \pm 0.02$ | $0.45 \pm 0.11$ |
| $M_D$ vs. $M'_{D2}$ ($D$) | $\mathbf{0.88} \pm 0.04$ | $0.52 \pm 0.01$ | $0.35 \pm 0.02$ | $0.42 \pm 0.06$ | $0.43 \pm 0.02$ | $0.52 \pm 0.03$ |
| $M_D$ vs. $M'_{D2}$ ($P$) | $\mathbf{0.90} \pm 0.04$ | $0.52 \pm 0.09$ | $0.45 \pm 0.07$ | $0.41 \pm 0.00$ | $0.44 \pm 0.01$ | $0.49 \pm 0.10$ |
| $M_C$ vs. $M'_{CN}$ ($G$) | $\mathbf{0.94} \pm 0.03$ | $0.50 \pm 0.02$ | $0.37 \pm 0.09$ | $0.39 \pm 0.13$ | $0.46 \pm 0.02$ | $0.52 \pm 0.21$ |
| $M_C$ vs. $M'_{CN}$ ($C$) | $\mathbf{0.89} \pm 0.02$ | $0.51 \pm 0.02$ | $0.28 \pm 0.12$ | $0.29 \pm 0.02$ | $0.39 \pm 0.04$ | $0.53 \pm 0.07$ |
| $M_C$ vs. $M'_{CN}$ ($MP$) | $\mathbf{0.95} \pm 0.01$ | $0.58 \pm 0.09$ | $0.31 \pm 0.07$ | $0.21 \pm 0.02$ | $0.30 \pm 0.00$ | $0.50 \pm 0.19$ |
| Average | $\mathbf{0.86} \pm 0.12$ | $0.61 \pm 0.23$ | $0.52 \pm 0.20$ | $0.54 \pm 0.18$ | $0.49 \pm 0.14$ | $0.47 \pm 0.15$ |

Table A4: **Concept Redundancy results across all comparison experiments.** We compute the Redundancy metric from SemanticLens [12]. The best (lowest) mean value per row is bolded (ties bolded). RDX has the best scores on the Redundancy metric.

| Comparison | RDX | KMeans | TKSAE | NMF | SAE | NLMCD |
|---|---|---|---|---|---|---|
| $M_{35}$ vs. $M_b$ | $0.99 \pm 0.00$ | $0.99 \pm 0.00$ | $0.99 \pm 0.00$ | $0.99 \pm 0.00$ | $\mathbf{0.98} \pm 0.00$ | $1.00 \pm 0.00$ |
| $M_{49}$ vs. $M_b$ | $0.99 \pm 0.00$ | $0.98 \pm 0.00$ | $0.99 \pm 0.01$ | $\mathbf{0.95} \pm 0.00$ | $0.97 \pm 0.00$ | $1.00 \pm 0.00$ |
| $M_{35}$ vs. $M_{49}$ | $0.96 \pm 0.02$ | $0.98 \pm 0.00$ | $0.99 \pm 0.00$ | $0.96 \pm 0.00$ | $0.99 \pm 0.00$ | $\mathbf{0.95} \pm 0.00$ |
| $M_{\leftrightarrow}$ vs. $M_{\rightleftarrows}$ | $\mathbf{0.95} \pm 0.00$ | $0.99 \pm 0.00$ | $0.98 \pm 0.00$ | $0.98 \pm 0.00$ | $0.99 \pm 0.00$ | $0.99 \pm 0.00$ |
| $M_{\updownarrow}$ vs. $M_{\uparrow\downarrow}$ | $\mathbf{0.97} \pm 0.01$ | $0.98 \pm 0.00$ | $0.99 \pm 0.00$ | $\mathbf{0.97} \pm 0.00$ | $0.98 \pm 0.00$ | $0.99 \pm 0.00$ |
| $C_{A-S}$ vs. $C_A$ | $0.87 \pm 0.01$ | $0.99 \pm 0.00$ | $\mathbf{0.86} \pm 0.00$ | $0.98 \pm 0.00$ | $0.92 \pm 0.01$ | $0.91 \pm 0.00$ |
| $C_{A-YB}$ vs. $C_A$ | $\mathbf{0.87} \pm 0.02$ | $1.00 \pm 0.00$ | $0.89 \pm 0.00$ | $0.98 \pm 0.00$ | $0.90 \pm 0.01$ | $0.90 \pm 0.00$ |
| $C_{A-YC}$ vs. $C_A$ | $\mathbf{0.81} \pm 0.02$ | $0.99 \pm 0.00$ | $0.87 \pm 0.00$ | $0.98 \pm 0.00$ | $0.90 \pm 0.00$ | $0.89 \pm 0.00$ |
| $C_{A-E}$ vs. $C_A$ | $\mathbf{0.84} \pm 0.01$ | $1.00 \pm 0.00$ | $0.88 \pm 0.01$ | $0.98 \pm 0.00$ | $0.92 \pm 0.00$ | $0.91 \pm 0.00$ |
| $C_{A-D}$ vs. $C_A$ | $\mathbf{0.80} \pm 0.05$ | $1.00 \pm 0.00$ | $0.85 \pm 0.00$ | $0.98 \pm 0.00$ | $0.90 \pm 0.00$ | $0.91 \pm 0.00$ |
| $M_D$ vs. $M_{D2}$ $(MT)$ | $0.80 \pm 0.02$ | $0.82 \pm 0.01$ | $0.78 \pm 0.00$ | $\mathbf{0.73} \pm 0.02$ | $0.81 \pm 0.00$ | $0.81 \pm 0.02$ |
| $M_D$ vs. $M_{D2}$ $(B)$ | $0.81 \pm 0.02$ | $0.86 \pm 0.00$ | $0.86 \pm 0.01$ | $0.84 \pm 0.02$ | $\mathbf{0.80} \pm 0.00$ | $0.91 \pm 0.00$ |
| $M_D$ vs. $M_{D2}$ $(D)$ | $\mathbf{0.76} \pm 0.00$ | $0.86 \pm 0.01$ | $0.80 \pm 0.01$ | $0.84 \pm 0.00$ | $0.85 \pm 0.00$ | $0.89 \pm 0.01$ |
| $M_D$ vs. $M_{D2}$ $(P)$ | $\mathbf{0.76} \pm 0.05$ | $0.87 \pm 0.00$ | $0.86 \pm 0.01$ | $0.80 \pm 0.00$ | $0.85 \pm 0.00$ | $0.89 \pm 0.00$ |
| $M_C$ vs. $M_{CN}$ $(G)$ | $0.84 \pm 0.02$ | $0.84 \pm 0.01$ | $0.81 \pm 0.01$ | $0.85 \pm 0.02$ | $0.84 \pm 0.00$ | $0.88 \pm 0.00$ |
| $M_C$ vs. $M_{CN}$ $(C)$ | $\mathbf{0.74} \pm 0.02$ | $0.81 \pm 0.02$ | $0.75 \pm 0.08$ | $0.85 \pm 0.01$ | $0.79 \pm 0.01$ | $0.86 \pm 0.00$ |
| $M_C$ vs. $M_{CN}$ $(MP)$ | $\mathbf{0.82} \pm 0.02$ | $0.88 \pm 0.01$ | $\mathbf{0.82} \pm 0.08$ | $0.84 \pm 0.02$ | $0.84 \pm 0.01$ | $0.89 \pm 0.00$ |
| $M_D$ vs. $M'_{D2}$ $(MT)$ | $\mathbf{0.79} \pm 0.04$ | $0.83 \pm 0.00$ | $0.80 \pm 0.01$ | $0.81 \pm 0.00$ | $0.84 \pm 0.02$ | $0.84 \pm 0.00$ |
| $M_D$ vs. $M'_{D2}$ $(B)$ | $0.83 \pm 0.02$ | $0.87 \pm 0.02$ | $\mathbf{0.83} \pm 0.01$ | $0.84 \pm 0.02$ | $0.85 \pm 0.01$ | $0.85 \pm 0.00$ |
| $M_D$ vs. $M'_{D2}$ $(D)$ | $\mathbf{0.79} \pm 0.00$ | $0.89 \pm 0.00$ | $0.82 \pm 0.02$ | $0.84 \pm 0.02$ | $0.87 \pm 0.00$ | $0.89 \pm 0.00$ |
| $M_D$ vs. $M'_{D2}$ $(P)$ | $\mathbf{0.80} \pm 0.03$ | $0.87 \pm 0.01$ | $0.88 \pm 0.00$ | $0.85 \pm 0.01$ | $0.85 \pm 0.01$ | $0.88 \pm 0.01$ |
| $M_C$ vs. $M'_{CN}$ $(G)$ | $\mathbf{0.82} \pm 0.01$ | $0.84 \pm 0.01$ | $0.83 \pm 0.01$ | $0.87 \pm 0.00$ | $0.86 \pm 0.01$ | $0.87 \pm 0.00$ |
| $M_C$ vs. $M'_{CN}$ $(C)$ | $\mathbf{0.80} \pm 0.00$ | $0.81 \pm 0.06$ | $0.84 \pm 0.03$ | $0.81 \pm 0.02$ | $\mathbf{0.80} \pm 0.00$ | $0.83 \pm 0.01$ |
| $M_C$ vs. $M'_{CN}$ $(MP)$ | $\mathbf{0.78} \pm 0.04$ | $0.88 \pm 0.06$ | $0.83 \pm 0.02$ | $0.84 \pm 0.06$ | $0.81 \pm 0.01$ | $0.89 \pm 0.00$ |
| Average | $\mathbf{0.84} \pm 0.07$ | $0.90 \pm 0.07$ | $0.87 \pm 0.06$ | $0.89 \pm 0.07$ | $0.87 \pm 0.06$ | $0.90 \pm 0.05$ |

Table A5: **Concept Clarity results across all comparison experiments.** We compute the Clarity metric from SemanticLens [12]. The best (highest) mean value per row is bolded (ties bolded). No method has a clear advantage on the Clarity metric.

| Comparison | RDX | KMeans | TKSAE | NMF | SAE | NLMCD |
|---|---|---|---|---|---|---|
| $M_{35}$ vs. $M_b$ | $0.94 \pm 0.04$ | $0.93 \pm 0.06$ | $0.94 \pm 0.03$ | $0.94 \pm 0.04$ | $0.90 \pm 0.08$ | $\mathbf{0.99} \pm 0.01$ |
| $M_{49}$ vs. $M_b$ | $0.96 \pm 0.03$ | $0.93 \pm 0.05$ | $0.91 \pm 0.06$ | $0.91 \pm 0.06$ | $0.94 \pm 0.09$ | $\mathbf{0.99} \pm 0.01$ |
| $M_{35}$ vs. $M_{49}$ | $0.95 \pm 0.02$ | $0.94 \pm 0.05$ | $0.95 \pm 0.04$ | $0.92 \pm 0.05$ | $0.92 \pm 0.02$ | $\mathbf{0.97} \pm 0.03$ |
| $M_{\leftrightarrow}$ vs. $M_{\rightleftarrows}$ | $\mathbf{0.92} \pm 0.06$ | $0.88 \pm 0.05$ | $0.87 \pm 0.04$ | $0.91 \pm 0.05$ | $0.91 \pm 0.04$ | $0.89 \pm 0.04$ |
| $M_{\updownarrow}$ vs. $M_{\uparrow\downarrow}$ | $\mathbf{0.94} \pm 0.03$ | $0.89 \pm 0.05$ | $0.89 \pm 0.02$ | $0.88 \pm 0.04$ | $0.89 \pm 0.04$ | $0.92 \pm 0.05$ |
| $C_{A-S}$ vs. $C_A$ | $0.43 \pm 0.04$ | $\mathbf{0.45} \pm 0.04$ | $0.44 \pm 0.03$ | $0.41 \pm 0.03$ | $\mathbf{0.45} \pm 0.06$ | $0.43 \pm 0.06$ |
| $C_{A-YB}$ vs. $C_A$ | $0.43 \pm 0.06$ | $\mathbf{0.45} \pm 0.04$ | $0.42 \pm 0.03$ | $0.42 \pm 0.03$ | $0.44 \pm 0.06$ | $\mathbf{0.45} \pm 0.09$ |
| $C_{A-YC}$ vs. $C_A$ | $\mathbf{0.48} \pm 0.06$ | $0.45 \pm 0.04$ | $0.43 \pm 0.03$ | $0.41 \pm 0.03$ | $0.43 \pm 0.05$ | $0.45 \pm 0.04$ |
| $C_{A-E}$ vs. $C_A$ | $0.43 \pm 0.02$ | $0.44 \pm 0.04$ | $0.42 \pm 0.02$ | $0.41 \pm 0.02$ | $\mathbf{0.45} \pm 0.06$ | $\mathbf{0.45} \pm 0.06$ |
| $C_{A-D}$ vs. $C_A$ | $\mathbf{0.46} \pm 0.08$ | $0.45 \pm 0.04$ | $0.44 \pm 0.03$ | $0.42 \pm 0.03$ | $0.44 \pm 0.05$ | $0.44 \pm 0.08$ |
| $M_D$ vs. $M_{D2}$ $(MT)$ | $0.40 \pm 0.06$ | $\mathbf{0.42} \pm 0.04$ | $0.38 \pm 0.05$ | $0.41 \pm 0.03$ | $0.36 \pm 0.04$ | $0.39 \pm 0.06$ |
| $M_D$ vs. $M_{D2}$ $(B)$ | $0.49 \pm 0.05$ | $\mathbf{0.50} \pm 0.06$ | $\mathbf{0.50} \pm 0.04$ | $0.48 \pm 0.08$ | $0.44 \pm 0.05$ | $0.45 \pm 0.05$ |
| $M_D$ vs. $M_{D2}$ $(D)$ | $0.45 \pm 0.05$ | $\mathbf{0.50} \pm 0.05$ | $0.45 \pm 0.07$ | $\mathbf{0.50} \pm 0.06$ | $0.42 \pm 0.06$ | $0.43 \pm 0.04$ |
| $M_D$ vs. $M_{D2}$ $(P)$ | $\mathbf{0.48} \pm 0.12$ | $0.47 \pm 0.06$ | $0.42 \pm 0.08$ | $0.46 \pm 0.12$ | $0.40 \pm 0.05$ | $0.45 \pm 0.09$ |
| $M_C$ vs. $M_{CN}$ $(G)$ | $0.41 \pm 0.06$ | $0.41 \pm 0.04$ | $\mathbf{0.42} \pm 0.04$ | $\mathbf{0.42} \pm 0.04$ | $0.36 \pm 0.07$ | $0.39 \pm 0.04$ |
| $M_C$ vs. $M_{CN}$ $(C)$ | $0.43 \pm 0.11$ | $\mathbf{0.46} \pm 0.11$ | $0.45 \pm 0.12$ | $\mathbf{0.46} \pm 0.08$ | $0.39 \pm 0.09$ | $0.41 \pm 0.07$ |
| $M_C$ vs. $M_{CN}$ $(MP)$ | $0.46 \pm 0.09$ | $\mathbf{0.50} \pm 0.08$ | $0.49 \pm 0.06$ | $0.47 \pm 0.07$ | $0.43 \pm 0.09$ | $0.44 \pm 0.07$ |
| $M_D$ vs. $M'_{D2}$ $(MT)$ | $0.40 \pm 0.04$ | $0.38 \pm 0.03$ | $\mathbf{0.42} \pm 0.06$ | $0.41 \pm 0.06$ | $0.37 \pm 0.03$ | $0.38 \pm 0.04$ |
| $M_D$ vs. $M'_{D2}$ $(B)$ | $0.44 \pm 0.05$ | $0.47 \pm 0.05$ | $\mathbf{0.49} \pm 0.05$ | $0.48 \pm 0.07$ | $0.46 \pm 0.06$ | $0.43 \pm 0.06$ |
| $M_D$ vs. $M'_{D2}$ $(D)$ | $0.44 \pm 0.04$ | $0.48 \pm 0.06$ | $0.48 \pm 0.06$ | $\mathbf{0.50} \pm 0.07$ | $0.48 \pm 0.06$ | $0.42 \pm 0.03$ |
| $M_D$ vs. $M'_{D2}$ $(P)$ | $0.46 \pm 0.07$ | $0.43 \pm 0.05$ | $\mathbf{0.48} \pm 0.04$ | $\mathbf{0.48} \pm 0.09$ | $0.42 \pm 0.05$ | $0.39 \pm 0.06$ |
| $M_C$ vs. $M'_{CN}$ $(G)$ | $0.40 \pm 0.06$ | $0.38 \pm 0.05$ | $0.41 \pm 0.02$ | $\mathbf{0.45} \pm 0.06$ | $0.39 \pm 0.06$ | $0.36 \pm 0.03$ |
| $M_C$ vs. $M'_{CN}$ $(C)$ | $0.41 \pm 0.05$ | $\mathbf{0.45} \pm 0.10$ | $0.42 \pm 0.09$ | $\mathbf{0.45} \pm 0.11$ | $0.40 \pm 0.10$ | $0.38 \pm 0.06$ |
| $M_C$ vs. $M'_{CN}$ $(MP)$ | $0.49 \pm 0.08$ | $\mathbf{0.52} \pm 0.08$ | $0.48 \pm 0.06$ | $0.50 \pm 0.09$ | $0.40 \pm 0.07$ | $0.38 \pm 0.06$ |
| Average | $\mathbf{0.55} \pm 0.21$ | $0.53 \pm 0.17$ | $0.52 \pm 0.18$ | $0.52 \pm 0.18$ | $0.49 \pm 0.19$ | $0.51 \pm 0.20$ |

Table A6: **Concept Polysemanticity results across all comparison experiments.** We compute the Polysemanticity metric from SemanticLens [12]. The best (lowest) mean value per row is bolded (ties bolded). No method has a clear advantage on the Polysemanticity metric.

| Comparison | RDX | KMeans | TKSAE | NMF | SAE | NLMCD |
|---|---|---|---|---|---|---|
| $M_{35}$ vs. $M_b$ | $0.08 \pm 0.11$ | $0.08 \pm 0.10$ | $0.07 \pm 0.05$ | $0.08 \pm 0.11$ | $0.12 \pm 0.13$ | $\mathbf{0.01} \pm 0.02$ |
| $M_{49}$ vs. $M_b$ | $0.09 \pm 0.07$ | $0.08 \pm 0.09$ | $0.06 \pm 0.04$ | $0.14 \pm 0.15$ | $0.09 \pm 0.14$ | $\mathbf{0.00} \pm 0.00$ |
| $M_{35}$ vs. $M_{49}$ | $0.07 \pm 0.04$ | $0.07 \pm 0.09$ | $0.06 \pm 0.04$ | $0.11 \pm 0.13$ | $0.09 \pm 0.07$ | $\mathbf{0.04} \pm 0.07$ |
| $M_{\leftrightarrow}$ vs. $M_{\rightleftarrows}$ | $\mathbf{0.06} \pm 0.05$ | $0.11 \pm 0.04$ | $0.12 \pm 0.05$ | $0.12 \pm 0.07$ | $0.11 \pm 0.07$ | $0.09 \pm 0.08$ |
| $M_{\updownarrow}$ vs. $M_{\uparrow\downarrow}$ | $0.11 \pm 0.06$ | $0.10 \pm 0.05$ | $0.10 \pm 0.03$ | $\mathbf{0.08} \pm 0.04$ | $0.14 \pm 0.08$ | $\mathbf{0.08} \pm 0.05$ |
| $C_{A-S}$ vs. $C_A$ | $0.38 \pm 0.06$ | $\mathbf{0.34} \pm 0.05$ | $0.38 \pm 0.07$ | $0.42 \pm 0.06$ | $0.35 \pm 0.08$ | $0.37 \pm 0.09$ |
| $C_{A-YB}$ vs. $C_A$ | $0.40 \pm 0.13$ | $\mathbf{0.34} \pm 0.05$ | $0.40 \pm 0.06$ | $0.40 \pm 0.07$ | $0.39 \pm 0.06$ | $\mathbf{0.34} \pm 0.10$ |
| $C_{A-YC}$ vs. $C_A$ | $\mathbf{0.33} \pm 0.10$ | $0.35 \pm 0.05$ | $0.40 \pm 0.09$ | $0.38 \pm 0.06$ | $0.36 \pm 0.06$ | $0.34 \pm 0.07$ |
| $C_{A-E}$ vs. $C_A$ | $0.35 \pm 0.07$ | $0.35 \pm 0.04$ | $0.38 \pm 0.06$ | $0.41 \pm 0.07$ | $\mathbf{0.34} \pm 0.07$ | $0.38 \pm 0.12$ |
| $C_{A-D}$ vs. $C_A$ | $\mathbf{0.27} \pm 0.10$ | $0.35 \pm 0.04$ | $0.39 \pm 0.08$ | $0.39 \pm 0.06$ | $0.38 \pm 0.08$ | $0.37 \pm 0.12$ |
| $M_D$ vs. $M_{D2}$ ($MT$) | $0.41 \pm 0.06$ | $0.39 \pm 0.07$ | $0.45 \pm 0.09$ | $0.40 \pm 0.07$ | $0.41 \pm 0.06$ | $\mathbf{0.38} \pm 0.10$ |
| $M_D$ vs. $M_{D2}$ ($B$) | $0.30 \pm 0.09$ | $\mathbf{0.29} \pm 0.08$ | $\mathbf{0.29} \pm 0.05$ | $0.32 \pm 0.09$ | $0.37 \pm 0.09$ | $0.33 \pm 0.07$ |
| $M_D$ vs. $M_{D2}$ ($D$) | $0.36 \pm 0.07$ | $\mathbf{0.31} \pm 0.05$ | $0.34 \pm 0.07$ | $0.32 \pm 0.08$ | $0.34 \pm 0.07$ | $0.36 \pm 0.07$ |
| $M_D$ vs. $M_{D2}$ ($P$) | $0.32 \pm 0.10$ | $\mathbf{0.30} \pm 0.07$ | $0.41 \pm 0.09$ | $0.34 \pm 0.11$ | $0.45 \pm 0.11$ | $0.39 \pm 0.12$ |
| $M_C$ vs. $M_{CN}$ ($G$) | $0.38 \pm 0.08$ | $0.41 \pm 0.06$ | $\mathbf{0.37} \pm 0.07$ | $\mathbf{0.37} \pm 0.09$ | $0.48 \pm 0.10$ | $0.43 \pm 0.10$ |
| $M_C$ vs. $M_{CN}$ ($C$) | $0.41 \pm 0.12$ | $0.39 \pm 0.12$ | $0.38 \pm 0.11$ | $\mathbf{0.38} \pm 0.12$ | $0.42 \pm 0.10$ | $0.43 \pm 0.08$ |
| $M_C$ vs. $M_{CN}$ ($MP$) | $0.39 \pm 0.12$ | $\mathbf{0.34} \pm 0.12$ | $\mathbf{0.34} \pm 0.09$ | $0.38 \pm 0.10$ | $0.40 \pm 0.09$ | $0.41 \pm 0.10$ |
| $M_D$ vs. $M'_{D2}$ ($MT$) | $0.39 \pm 0.07$ | $0.40 \pm 0.07$ | $\mathbf{0.38} \pm 0.10$ | $0.39 \pm 0.08$ | $0.40 \pm 0.05$ | $0.41 \pm 0.07$ |
| $M_D$ vs. $M'_{D2}$ ($B$) | $0.35 \pm 0.06$ | $\mathbf{0.30} \pm 0.06$ | $0.36 \pm 0.07$ | $0.33 \pm 0.09$ | $0.34 \pm 0.08$ | $0.36 \pm 0.06$ |
| $M_D$ vs. $M'_{D2}$ ($D$) | $0.36 \pm 0.09$ | $\mathbf{0.32} \pm 0.08$ | $\mathbf{0.32} \pm 0.07$ | $0.34 \pm 0.09$ | $0.36 \pm 0.10$ | $0.39 \pm 0.09$ |
| $M_D$ vs. $M'_{D2}$ ($P$) | $\mathbf{0.33} \pm 0.06$ | $0.38 \pm 0.08$ | $\mathbf{0.33} \pm 0.09$ | $0.34 \pm 0.09$ | $0.37 \pm 0.08$ | $0.44 \pm 0.14$ |
| $M_C$ vs. $M'_{CN}$ ($G$) | $0.42 \pm 0.11$ | $0.46 \pm 0.07$ | $0.43 \pm 0.05$ | $\mathbf{0.38} \pm 0.08$ | $0.41 \pm 0.09$ | $0.47 \pm 0.11$ |
| $M_C$ vs. $M'_{CN}$ ($C$) | $0.44 \pm 0.09$ | $0.39 \pm 0.13$ | $\mathbf{0.38} \pm 0.11$ | $\mathbf{0.38} \pm 0.18$ | $0.44 \pm 0.11$ | $0.51 \pm 0.15$ |
| $M_C$ vs. $M'_{CN}$ ($MP$) | $0.33 \pm 0.11$ | $0.31 \pm 0.13$ | $0.33 \pm 0.06$ | $\mathbf{0.30} \pm 0.07$ | $0.43 \pm 0.08$ | $0.43 \pm 0.12$ |
| Average | $\mathbf{0.31} \pm 0.12$ | $\mathbf{0.31} \pm 0.11$ | $0.33 \pm 0.11$ | $0.33 \pm 0.11$ | $0.35 \pm 0.12$ | $0.34 \pm 0.13$ |

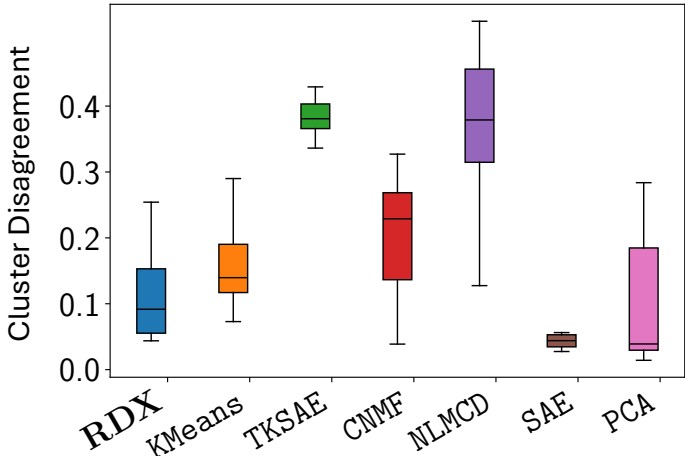

Figure A11: **Cluster Disagreement.** We report how much concepts change under in-distribution dataset perturbations (see Appendix A.5). On the y-axis we plot the cluster disagreement, which measures how much clusters disagree under in-distribution perturbations of the dataset. Lower cluster disagreement indicates a higher concept consistency. RDX has fairly low disagreement, indicating it generates reasonably consistent concepts under dataset perturbations.

## A.5 Concept Consistency Under In-Distribution Dataset Perturbations

Our aim in this section is to test whether concepts remain the same when there are *in-distribution* changes to the data. While concept shifts are expected when the distribution changes, a desirable property of concept extraction methods is *consistency* when the inputs are changed, but the underlying distribution remains the same, e.g., uniformly adding or removing images from a class to see if it is has any impact on the identified concept differences.

To test this, we use four ImageNet-derived base datasets: *Primates*, *Mittens*, *Buses*, and *Dogs* (details in Table A11). For each base dataset, we create two in-distribution variants by (i) uniformly removing 20% of datapoints (-20%) and (ii) uniformly adding 20% more datapoints sampled from the same classes as the base dataset (+20%). This yields $4 \times 3 = 12$ datasets: {base, -20%, +20%} for each of the four bases. On each of the 12 datasets, we extract concepts for the DINO [6] vs. DINOv2 [43] comparison using seven methods: RDX, KMeans, TKSAE, CNMF, NLMCD, SAE, and PCA. For the present analysis, however, our focus is *not* on cross-model differences; instead, we assess how consistent the extracted concepts are *within* a given base dataset across its in-distribution variants (base, $-20\%$, $+20\%$).

To measure consistency, we would like to compare how similar concepts are across the different dataset variants. We represent each concept via a set of images that contain it. Specifically, we convert each method's output into a hard clustering over images, where each cluster corresponds to a concept. For RDX, KMeans, and NLMCD we use their native hard cluster assignments. For TKSAE, CNMF, SAE and PCA, which yield soft assignments, we assign each image to the concept with the largest coefficient (argmax over concept activations).

For each base dataset, we compare clusterings across the three in-distribution variants using the pairs: (base, -20%), (base, +20%), and (-20%, +20%). To make the comparisons feasible, we restrict them to the intersection of images that are present in all three variants; after filtering, each dataset contains 1200 images. We align clusters between dataset variants using the Hungarian algorithm [31] and report the *cluster disagreement*: the fraction of images whose assigned clusters disagree after alignment. Lower values indicate higher concept consistency.

Fig. A11 summarizes the disagreement across all bases and pairs. SAE shows the highest consistency, exhibiting almost no disagreement across dataset variations. RDX performs well, maintaining low disagreement under in-distribution perturbations. In contrast, NLMCD, TKSAE and CNMF display larger disagreement scores, indicating less stability. KMeans and PCA remain relatively consistent as well.

## A.6    Discussion of Euclidean Distances

Euclidean distances can sometimes be misleading in neural network representations (see Sec. 5). In this section we discuss why RDX can still work despite this issue.

**Many models have meaningful Euclidean distances in the final layer.** Many modern self-supervised models, like DINOv2 [43], have meaningful Euclidean distances in the latent space as seen by their performance on KNN probing and linear probing (Table 4 in [43]). Additionally, the training recipe for most classification models is likely to result in *relatively* meaningful Euclidean distances. Classification models are commonly trained with a final linear layer and a cross-entropy loss. The final layer outputs logits that are converted to probabilities using a function like softmax. These logits are not guaranteed to have meaningful Euclidean distances, since scaling and translation do not affect the softmax function, however, in order to minimize the cross-entropy loss they must have meaningful *relative* Euclidean distances. In our experiments, we use the final layer in all experiments.

**Impact of using the *nearest* neighbors ranking.** It is unlikely that the nearest neighbors for a given point have small Euclidean distances, but large distances on the manifold, since we generally expect that representations are *locally* linear. Neighborhood distances and our locally-biased difference function push RDX to seek out groups of points that are locally linear. Notably, the non-linear dimensionality reduction technique Isomap [61] uses top-k nearest neighbors to construct a graph and estimate distances along the manifold of the data. Our locally-biased difference function is a soft version of the same idea.

## B    Additional Methods Description

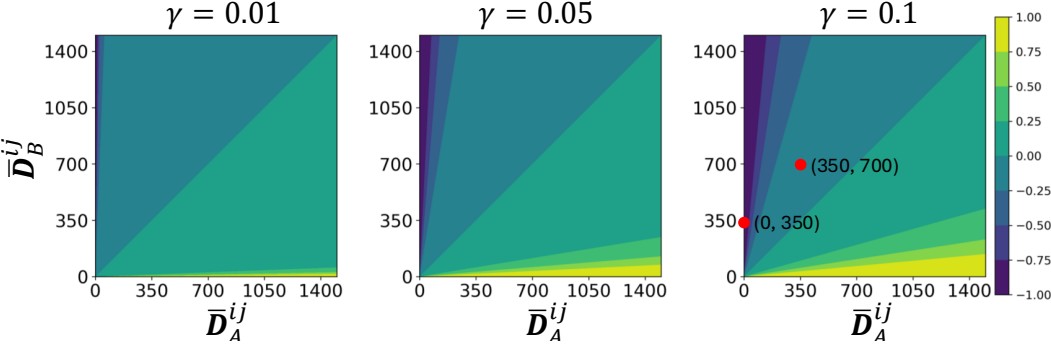

Figure A12: **Locally Biased Difference Function.** Here we visualize the locally biased difference function described in Sec. 3.2. On the x-axis we plot $\bar{\boldsymbol{D}}_A^{ij}$, which is the normalized distance in $\boldsymbol{A}$. On the y-axis we plot $\bar{\boldsymbol{D}}_B^{ij}$. Each panel corresponds to a different $\gamma$ value. We analyze the $\gamma = 0.1$ sub-panel in more detail. When $\bar{\boldsymbol{D}}_A^{ij}$ is smaller than the distance in $\bar{\boldsymbol{D}}_B^{ij}$ the difference value is negative, indicating by darker colors in the top left triangle half. However, we can see that the same relative change does not result in the same difference value. We mark two points in red on the plot. Both points represent the same amount of change when comparing the distance matrices, $\bar{\boldsymbol{D}}_B^{ij} - \bar{\boldsymbol{D}}_A^{ij} = 350$. However, the difference functions considers the change from 0 to 350 as more important (returns -1), whereas the change from 350 to 750 is less important and (returns around -0.25). Thus, the difference function reacts most strongly when highly similar items in $\boldsymbol{A}$ are considered less similar in $\boldsymbol{B}$. The $\gamma$ value adjusts how quickly the function saturates.

### B.1    K-neighborhood Affinity (KNA) Pseudocode

Let $\boldsymbol{F}_{A,B} \in \mathbb{R}^{n \times n}$ denote the full affinity matrix computed between all image pairs using representations $\boldsymbol{A}$ and $\boldsymbol{B}$. For a given spectral cluster $C \subseteq \{1, \ldots, n\}$, we extract the submatrix $\boldsymbol{F}_{A,B}^C \in \mathbb{R}^{r \times r}$, where $r = |C|$, by selecting only the rows and columns of $\boldsymbol{F}_{A,B}$

corresponding to the indices in $C$. This subset of the affinity matrix $\boldsymbol{F}_{A,B}^C$ captures pairwise affinities within the cluster and serves as input to the KNA-based selection procedure.

---

**Algorithm 1** Selecting Image and Neighbors with Maximum KNA

---

**Require:** Cluster $C = \{x_1, x_2, \ldots, x_r\}$, Subset of affinity matrix $\boldsymbol{F}_{A,B}^C \in \mathbb{R}^{r \times r}$, Number of neighbors $k$
**Ensure:** Image $x_{\max}$ and its $k$-nearest neighbors $N_{\max}$
1: **for** each image $x_i$ in $C$ **do**
2:     $N_i \leftarrow$ indices of the $k$ largest values in row $A[i,:]$
3:     $\text{KNA}(x_i) \leftarrow \sum_{j \in N_i} A[i,j]$
4: **end for**
5: $x_{\max} \leftarrow \arg\max_{x_i} \text{KNA}(x_i)$
6: $N_{\max} \leftarrow N_{x_{\max}}$
7: **return** $x_{\max}, N_{\max}$

---

## B.2 Computing SemanticLens metrics

We compute the Redundancy, Clarity, and Polysemanticity metrics from SemanticLens [12]. These metrics were originally designed for analyzing concept explanations for a single model. We measure Clarity and Polysemanticity for each concept as described in the original work. However, we modify the Redundancy metric to better analyze concepts in the context of model comparison. Rather than compute similarity between all concepts, we measure the similarity between concepts from one model to the concepts of the other model.

Following SemanticLens, we use a generalist model, OpenCLIP [23], to analyze each concept $k$ via its explanation grid $E_k$ (see Sec. 3.3). We refer to the OpenCLIP image encoder as $\mathcal{F}$. Let $E_k = \{x_{k,i}\}_{i=1}^{n_k}$ denote the set of images in the explanation grid for concept $k$. The following equations are the same as those presented in SemanticLens [12], with minor changes to match the notation used in this work.

For each image $x_{k,i} \in E_k$, we obtain its OpenCLIP image embedding via

$$v_{k,i} \;=\; \mathcal{F}(x_{k,i}) \in \mathbb{R}^d, \tag{3}$$

and collect them into

$$V_k \;=\; \begin{bmatrix} v_{k,1} \; v_{k,2} \; \cdots \; v_{k,n_k} \end{bmatrix}^\top \in \mathbb{R}^{n_k \times d}. \tag{4}$$

### B.2.1 Clarity

The Clarity of a concept $k$ is computed using the embeddings $(V_k)$ of the images in its explanation $(E_k)$. The more similar the images within an explanation are to each other, the clearer the explanation is. Cosine similarity, denoted $s_{cos}$, is used to measure the similarity between images.

$$I_{\texttt{clarity}}(V_k) := \frac{1}{|V_k|(|V_k|-1)} \sum_{i=1}^{|V_k|} \sum_{j \neq i} s_{\cos}(\mathbf{v}_{k,i}, \mathbf{v}_{k,j}) \tag{5}$$

$$= \frac{|V_k|}{|V_k|-1} \left( \left\| \frac{1}{|V_k|} \sum_{i=1}^{|V_k|} \frac{\mathbf{v}_{k,i}}{\|\mathbf{v}_{k,i}\|_2} \right\|_2^2 - \frac{1}{|V_k|} \right) \in \left[ -\frac{1}{|V_k|-1}, 1 \right] \tag{6}$$

### B.2.2 Polysemanticity

Polysemantic concepts are concepts that can be decomposed into two or more simpler concepts. In SemanticLens, concepts are considered polysemantic if subsets of $E_k$ can be identified that result in diverging semantic directions. The metric is defined as

$$I_{\texttt{poly}}(V_k^{(1)}, \ldots, V_k^{(h)}) := 1 - I_{\texttt{clarity}} \left( \left\{ \sum_{\mathbf{v} \in V_k^{(i)}} \mathbf{v} \mid i = 1, \ldots, h \right\} \right), \tag{7}$$

where $V_k^{(i)} \subseteq V_k$ for $i = 1, ..., h$ is a subset of the embedded explanation images. Subsets are discovered using KMeans with two clusters.

### B.2.3 Redundancy

In the context of model comparison, we define redundancy as the similarity between concepts derived from two models. Let $A$ and $B$ denote the models, and let $V_k^A$ and $V_l^B$ represent the set of OpenCLIP embeddings corresponding to their respective concepts. In the following equations, we use the *mean concept embedding*, denoted by $\bar{V}_k^A$, to represent the mean embedding of the $k^{\text{th}}$ concept extracted from model $A$. It is computed as:

$$\bar{V}_k^A = \frac{1}{n_k} \sum_{i=1}^{n_k} V_{k,i}^A, \tag{8}$$

where $n_k$ denotes the number of images assigned to the explanation of the $k^{th}$ concept and $V_{k,i}^A$ is the embedding of the $i^{th}$ image belonging to that concept. We compute the cosine similarity between mean concept embeddings as follows:

$$I_{\texttt{sim}}(\bar{V}_k^A, \bar{V}_l^B) := s_{\cos}(\bar{V}_k^A, \bar{V}_l^B) = \frac{\langle \bar{V}_k^A, \bar{V}_l^B \rangle}{\|\bar{V}_k^A\|_2 \|\bar{V}_l^B\|_2} \in [-1, 1], \tag{9}$$

We compute this value for each pair of concepts across the two models, resulting in a $k \times l$ similarity matrix. To summarize this information into a single score, we compute the mean of the maximum similarity values per concept, which we refer to as the *redundancy score*. The maximum similarity for a concept is low when there is no similar concept from the other model. Consequently, a low redundancy score indicates that the concepts from the two models are less similar, a desirable property when seeking concepts that are unique to each model.

$$I_{\texttt{red}}(\mathcal{V}^A, \mathcal{V}^B) := \frac{1}{|\mathcal{V}^A|} \sum_{k=1}^{|\mathcal{V}^A|} \max_j \left\{ I_{\texttt{sim}}(\bar{V}_k^A, \bar{V}_l^B) \right\} \in [-1, 1], \tag{10}$$

To obtain a symmetric measure of redundancy, we average the scores in both directions:

$$I_{\texttt{red}}^{\text{sym}}(\mathcal{V}^A, \mathcal{V}^B) := \frac{1}{2} \left[ I_{\texttt{red}}(\mathcal{V}^A, \mathcal{V}^B) + I_{\texttt{red}}(\mathcal{V}^B, \mathcal{V}^A) \right]. \tag{11}$$

### B.3 Normalized Distance Variants

Here we describe two alternatives to neighborhood distances. Both variants start by computing the pairwise Euclidean distance for each embedding matrix, resulting in $\boldsymbol{D}_A \in R^{n \times n}$ and $\boldsymbol{D}_B \in R^{n \times n}$.

**Max-normalized Euclidean Distances.** Each distance matrix is divided by the maximum distance in the matrix, such that both $\boldsymbol{D}_A$ and $\boldsymbol{D}_B$ are normalized between 0 and 1. Referred to as $\texttt{RDX}_{MN}$.

**Locally Scaled Euclidean Distances.** We compute a locally-scaled Euclidean distance that has been shown to have desirable properties for clustering [71]. For each embedding vector $\boldsymbol{a}_i$, this function scales the latent distances between $\boldsymbol{a}_i$ and all other inputs $\boldsymbol{a}_j$ by the distance $\boldsymbol{D}_A^{ik}$, where $\boldsymbol{a}_k$ is the 7th neighbor of $\boldsymbol{a}_i$. Referred to as $\texttt{RDX}_{LS}$.

**BSR Variants.** We also use these variants to compute the $\texttt{BSR}$ metric. We refer to the variants as $\texttt{BSR}_{MN}$ and $\texttt{BSR}_{LS}$. $\texttt{BSR}$ with no subscript uses neighborhood distances.

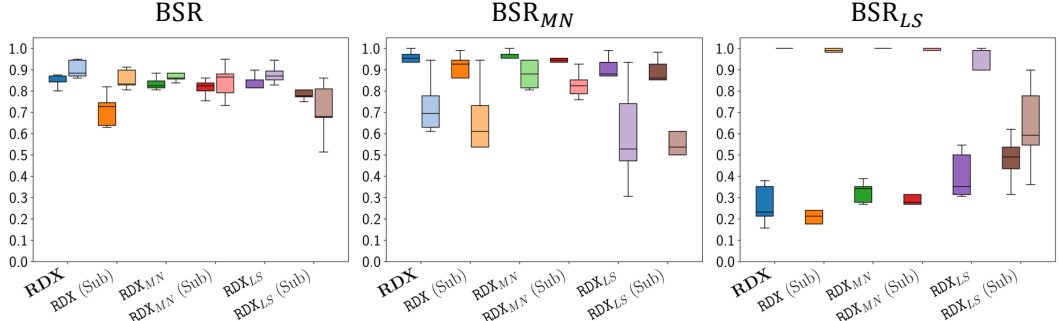

Figure A13: **BSR variants for RDX difference function variants.** We compute BSR variants for all RDX distance variants using each difference function. By default we use the locally biased difference function, we denote experiments with the subtraction difference function as RDX (Sub). We compute the BSR metric on the MNIST experiments. We see, that $BSR_{LS}$ is a poor metric, thus we focus on BSR (neighborhood) and $BSR_{MN}$ to assess the different variants. See Appendix B.6 for a longer discussion. Under all distance variants, we can see that RDX with the locally biased difference function outperforms subtraction.

## B.4 Difference Matrix Function Variants

**Subtraction.** The simplest approach to comparing the normalized distance matrices is subtraction:

$$\boldsymbol{G}_{A,B} = \bar{\boldsymbol{D}}_A - \bar{\boldsymbol{D}}_B. \tag{12}$$

If the distance between two inputs is small in $\bar{\boldsymbol{D}}_A$ and large in $\bar{\boldsymbol{D}}_B$, it would result in a large negative value in the difference matrix. If the distances are approximately equal in both matrices, then it would result in a value near zero in the difference matrix. Therefore, images considered similar in only one of the two representations would be identified by large negative values in $\boldsymbol{G}_{A,B}^{ij}$. Unfortunately, subtraction can be sensitive to imperfect normalization and/or large changes in already distant embeddings.

## B.5 Difference Explanation Sampling Variants

**PageRank.** We rank nodes in the graph by their PageRank [44]. Let the node with the largest PageRank be $i$. We select the $|E| - 1$ nodes corresponding to the $|E| - 1$ largest edges with one endpoint at $i$. We remove these nodes from the pool and iterate until all $m$ sets of explanation grids ($\mathcal{E}$) are sampled.

## B.6 Results

In Fig. A13 and Fig. A14 we evaluate the different variants for RDX. We find that all RDX variants perform better than baseline methods indicating that using both representations to isolate differences is an effective strategy.

First, when comparing difference functions on known MNIST comparisons (Fig. A13), we see a consistent advantage for the locally biased difference function. In all other experiments, we use the locally biased difference function. Second, PageRank [44] sampling is slightly worse than our spectral cluster and sample approach (Fig. A14). Third, we notice that $BSR_{LS}$ is a flawed metric. One comparison direction consistently scores near perfectly while the other is quite poor, indicating that distances are not comparable across the two representations. This indicates that local scaling [71], is not appropriate when comparing across representations, although future work may be able to modify it appropriately. Finally, when comparing RDX to $RDX_{MN}$ we notice that they perform reasonably similarly in the metrics. In Table A7, we show results for a comparison between $\boldsymbol{M}_{35}$ and $\boldsymbol{M}_b$ under BSR and $BSR_{MN}$ We can see that RDX with neighborhood distances performs well under BSR, but worse on $BSR_{MN}$. In contrast, $RDX_{MN}$ performs well on both metrics. We visualize the explanations for these methods in Fig. A15. While both methods have good explanations for $\boldsymbol{M}_{35}$ vs. $\boldsymbol{M}_b$, we can

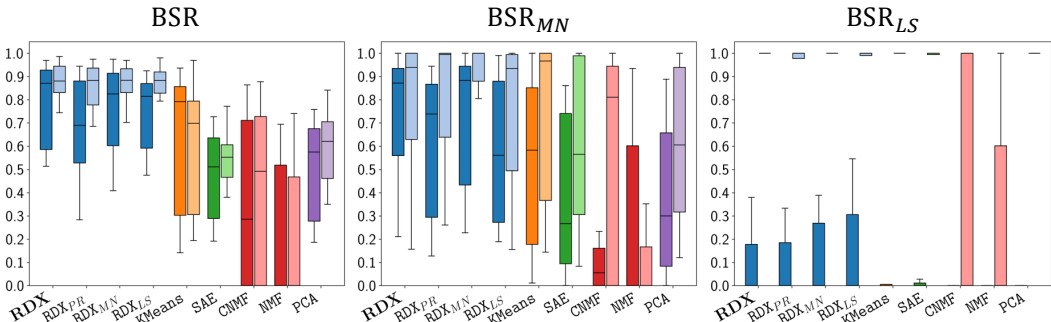

Figure A14: **BSR variants for RDX variants and baselines.** We compute BSR variants for several methods. We evaluate RDX variants with neighborhood distances (RDX), neighborhood distances and PageRank [44] sampling (RDX$_{PR}$), max-normalized distances (RDX$_{MN}$), and locally-scaled distances [71] (RDX$_{LS}$). We compute the BSR metric with all three distance function variants on the MNIST, CUB, and ImageNet/iNaturalist experiments (without alignment). We see that BSR$_{LS}$ is a poor metric as all methods perform perfectly in one of the two comparison directions, suggesting that the scaling technique does not make distances across representations comparable. We focus on BSR (neighborhood) and BSR$_{MN}$ to assess the different variants. First, we see broadly that RDX variants outperform all baseline methods. Among them, RDX and RDX$_{MN}$ perform the best, although RDX$_{MN}$ shows slightly greater variance.

see that in the other direction the two methods differ significantly. RDX with neighborhood distances is much more focused on the known difference than RDX$_{MN}$. This is likely due to the issue described in Sec. 3.2. Thus, we use RDX and BSR with neighborhood distances for the main experiments.

Table A7: Comparing RDX variants on $\boldsymbol{M}_{35}$ vs. $\boldsymbol{M}_b$ under different BSR variants.

|  | RDX($\boldsymbol{M}_{35}, \boldsymbol{M}_b$) | RDX$_{MN}$($\boldsymbol{M}_{35}, \boldsymbol{M}_b$) | RDX($\boldsymbol{M}_b, \boldsymbol{M}_{35}$) | RDX$_{MN}$($\boldsymbol{M}_b, \boldsymbol{M}_{35}$) |
|---|---|---|---|---|
| BSR | 0.80 | 0.86 | 0.82 | 0.88 |
| BSR$_{MN}$ | 0.95 | 0.63 | 0.97 | 0.81 |

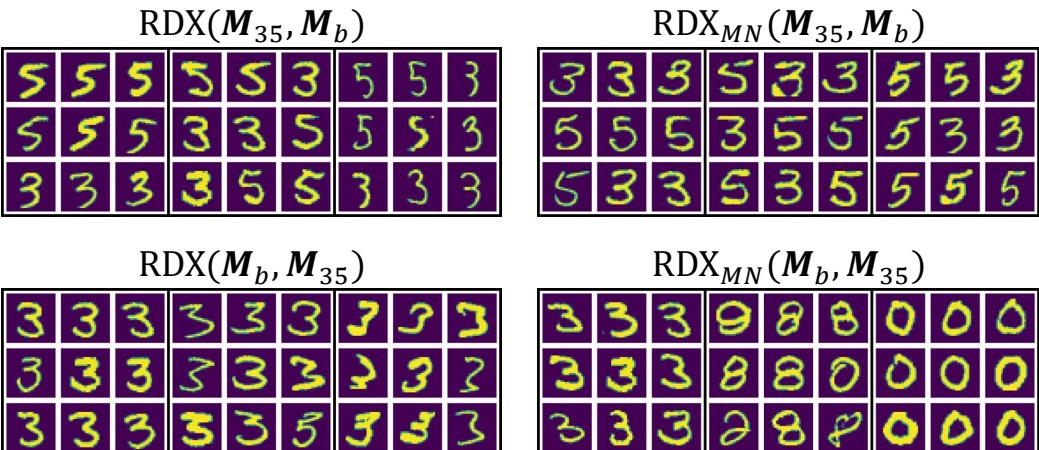

Figure A15: **Comparing explanations using max-normalized distances vs. neighborhood distances.** (Row 1) Both RDX variants generate good difference explanations that capture mixing that is unique to $M_{35}$. (Row 2) RDX with neighborhood distances focuses much more on the known difference with all three explanations showing cleanly grouped 3s. In contrast, $\text{RDX}_{MN}$ shows one group of 3s and two other groups unrelated to the known difference.

# C  Implementation Details

Table A8: Expected effects of "known" differences.

| Repr. ID | Modification | Expectation |
|---|---|---|
| $M_S$ | MNIST dataset only contains 3, 5, and 8. Model checkpoint is from epoch 1, step 184 with 94% | Mistakes on more challenging images. Clusters have slight overlaps. |
| $M_E$ | MNIST dataset only contains 3, 5, and 8. Final model checkpoint with 98% | Clusters have little to no overlap. |
| $M_b$ | None | Baseline model with well-separated clusters. |
| $M_{35}$ | Labels for 5 are replaced with 3. | 3s and 5s are mixed together in the representation. |
| $M_{49}$ | Labels for 9 are replaced with 4. | 4s and 9s are mixed together in the representation. |
| $M_{\leftrightarrow}$ | Dataset includes horizontally flipped images and uses the original label for flipped images. | Will mix flipped and unflipped digits together. |
| $M_{\rightleftarrows}$ | Dataset includes horizontally flipped images and uses new labels for flipped images. | Horizontally flipped digits are separated into new clusters. |
| $M_{\updownarrow}$ | Dataset includes vertically flipped images and uses the original label for flipped images. | Will mix flipped and unflipped digits together. |
| $M_{\uparrow\downarrow}$ | Dataset includes vertically flipped images and uses new labels for flipped images. | Vertically flipped digits are separated into new clusters. |
| $C_A$ | None | Baseline model with organized representational geometry. |
| $C_{A-S}$ | Remove the spotted wing concept. | Representational changes for images of birds with spotted wings. |
| $C_{A-YB}$ | Remove the yellow back concept. | Representational changes for images of birds with yellow backs. |
| $C_{A-YC}$ | Remove the yellow crown concept. | Representational changes for images of birds with yellow crowns. |
| $C_{A-E}$ | Remove the eyebrow on head concept. | Representational changes for images of birds with head eyebrows. |
| $C_{A-D}$ | Remove the duck-like shape concept. | Representational changes for images of birds with duck-like shapes. |

## C.1  Model Training

**Failures of Existing Methods on MNIST-[3,5,8]** (Sec. 4.2). We train a 2-layer convolutional network with an output dimension of eight on a modified MNIST [32] dataset that only contains the digits 3, 5, and 8 ((MNIST-[3,5,8]). The network is trained for five epochs with a batch size of 128. We use the Adam [27] optimizer with the learning rate set to 1e-2 and a one-cycle learning rate schedule. The global seed is set to 4834586. For the

comparison experiment, we select a checkpoint at epoch 1, step 184 with strong performance ($\boldsymbol{M}_S$, 95%) and the final checkpoint at epoch 5 with expert performance ($\boldsymbol{M}_E$, 98%).

**Recovering "Known" Differences** (Sec. 4.3). First, we train a 2-layer convolutional network with an output dimension of 64 on several modified MNIST datasets. See Table A8 for modification details. The network is trained for five epochs with a batch size of 128. We use the Adam [27] optimizer with the learning rate set to 1e-2 and a one-cycle learning rate schedule. The global seed is set to 4834586 for all models. Models are evaluated on the modified dataset that they were trained on. Second, we train a post-hoc concept bottleneck model (PCBM) [70] on the CUB dataset [68] using the original procedure [70]. The model backbone is a ResNet-18 [20] pre-trained on CUB from pytorchcv [60]. The concept classifier is from scikit-learn [47] and is trained with stochastic gradient descent with the elastic-net penalty. The learning rate is set to 1e-3 and the model is trained for a maximum of 10000 iterations with a batch size of 64. For the comparison experiments, we eliminate a concept by deleting the corresponding concept index from the predicted concept vector. The eliminated concepts are provided in Table A8. Models are compared on all images in the CUB train set.

Table A9: Models and their identifiers from the TIMM library.

| Repr. ID | Description | Timm Library ID |
|:---:|:---:|:---|
| $\boldsymbol{M}_D$ | DINO | `vit_base_patch16_224.dino` |
| $\boldsymbol{M}_{D2}$ | DINOv2 | `vit_base_patch14_reg4_dinov2.lvd142m` |
| $\boldsymbol{M}_C$ | CLIP | `hf_hub:timm/vit_large_patch14_clip_336.`
`openai` |
| $\boldsymbol{M}_{CN}$ | CLIP ft. iNat | `hf_hub:timm/vit_large_patch14_clip_336.`
`laion2b_ft_in12k_in1k_inat21` |

**Discovering "Unknown" Differences** (Sec. 4.4). All models in this experiment were downloaded from the timm library [64]. Details are available in Table A9.

## C.2 Alignment Training

To align representation $\boldsymbol{A}$ to representation $\boldsymbol{B}$, we learn a transformation matrix $\boldsymbol{M}_{AB} \in \mathbb{R}^{d_A \times d_A}$. We randomly sample 70% of the embeddings in our dataset to train the transformation matrix. The other 30% are used as a validation set. The matrix is trained for 100 steps, with the Adam optimizer [27] with a learning rate of 0.001. We measure the CKA on the validation set and keep the best transformation matrix.

## C.3 Baselines

For the baseline methods we use the scikit-learn [47] implementations for `PCA`, `NMF`, and `KMeans`. For `CNMF`, we use the pymf [63] implementation. For `TKSAE` (top-k sparse auto-encoder), we use the Overcomplete repository [25]. Top-k SAEs use an overcomplete basis of concepts, but zero all concept coefficients below the top-k coefficient values. In our experiments, we use a basis of 50 concepts and set $k = 3$ or $k = 5$ depending on the experiment. The encoder for the `TKSAE` is a linear layer with batch normalization. It is trained for 200 epochs with a batch size of 1024 or the maximum number of images. We use a linear learning rate warmup over the first 10 epochs, after which the learning rate is fixed at 0.0005. The model is trained with the Adam [27] optimizer. The sparsity coefficient is set to 0.0004. After training, `TKSAE` generates 50 concepts per model. In this work, each method is evaluated on $k$ concepts per model. To fairly compare `TKSAEs`, we only generate explanations for the $k$ concepts with the largest average coefficient. Note that coefficient values for each concept are directly comparable because the magnitude of each concept vectors is $l_2$-normalized to 1. For `SAEs`, we use code that is adapted from [58]. The `SAE` has a linear encoder, a relu activation and a linear decoder. Inputs are z-score normalized. It is trained for 500 epochs with a batch size of 2000 or the maximum number of images. The

dimension is set to the number of desired explanations (3 or 5 depending on the experiment). We use a linear learning rate warmup over the first 10 epochs, after which the learning rate is fixed at 0.001. The model is trained with the Adam [27] optimizer. The sparsity coefficient is set to 0.0004. For `NLMCD` [65] we use the publicly available official implementation. We modify this implementation to accept a pre-computed embedding matrix for each model, allowing us to use the same input embeddings for all methods. We use the default settings from the original paper to generate clusters using HDBSCAN [39]. This results in an unspecified number of clusters for each model. For a fair evaluation of `NLMCD`, we keep the $k$ clusters from $A$ that had the lowest similarity score to clusters from $B$, since we are interested in finding concepts unique to one of the two models.

For `PCA`, `NMF`, `CNMF`, `SAE` and `TKSAE` we generate explanations by sampling the $|E|$ images with the largest coefficients for each concept vector. For `KMeans`, we sample images closest to the centroid of the cluster. For `NLMCD`, we follow the methodology proposed in the original work and randomly sample $|E|$ images from each cluster.

## C.4 `RDX` Details

We sweep $\gamma$ on one comparison from each experiment group (see breaks in Table A11) and select the value that results in the highest performance on `BSR`. We find that a $\gamma$ of 0.05 or 0.1 works well. We set $\beta$ to 5 in all experiments.

## C.5 Comparison Summary

A complete list of comparisons, the data used in the comparison, and the number of images is available in Table A11. We choose to generate 3 explanations for all MNIST comparisons and 5 explanations for all other comparisons. We choose 3 or 5 because we prefer a small set of explanations for users to analyze. For all experiments, we use $3 \times 3$ image grids. In all comparisons where images are from an existing dataset, we use images from the train split because the train split is usually larger. Note that our method is training free and is not impacted by the dataset splits. For the iNaturalist comparisons, 600 research grade images are downloaded from the iNaturalist website [24] with licenses (cc-by,cc-by-nc,cc0). Images are restricted to be a maximum of 500 pixels on the longest side.

## C.6 Computational Cost

All experiments were conducted using on a machine with an AMD Ryzen 7 3700X 8-Core Processor and a single GeForce RTX 4090 GPU with 128GB of RAM. In Table A10, we show the time taken for each method on the CUB dataset (5000 images). $\text{RDX}_{PR}$ uses PageRank [44] to rank nodes for sampling and is slower. The time for `SAE` varies with the model's output dimension and the number of images. In this table, the model backbone is a ResNet18 [20] and has an output dimension of 512.

Table A10: Runtime (in seconds) for selected XAI methods.

|          | RDX   | $\text{RDX}_{PR}$ | KMeans | CNMF  | SAE    | PCA  | Classifiers |
|----------|-------|-------------------|--------|-------|--------|------|-------------|
| Time (s) | 32.71 | 187.78            | 9.65   | 14.95 | 629.95 | 8.49 | 97.3        |

Table A11: Experimental settings where we report comparison name, dataset, number of images, concepts, and gamma values. We name the comparisons using one direction, but compare in both directions in all experiments.

| Comparison | Comparison Dataset | Num. Ims. | $|\mathcal{E}|$ | RDX-$\gamma$ |
|:---:|:---:|:---:|:---:|:---:|
| $\boldsymbol{M}_S$ vs. $\boldsymbol{M}_E$ | MNIST-[3,5,8] | 500 x 3 | 3 | 0.05 |
| $\boldsymbol{M}_{35}$ vs. $\boldsymbol{M}_b$ | MNIST | 500 x 10 | 3 | 0.1 |
| $\boldsymbol{M}_{49}$ vs. $\boldsymbol{M}_b$ | MNIST | 500 x 10 | 3 | 0.1 |
| $\boldsymbol{M}_{35}$ vs. $\boldsymbol{M}_{49}$ | MNIST | 500 x 10 | 3 | 0.1 |
| $\boldsymbol{M}_{\leftrightarrow}$ vs. $\boldsymbol{M}_{\rightleftarrows}$ | MNIST w/ hflip | 250 x 20 | 3 | 0.1 |
| $\boldsymbol{M}_{\updownarrow}$ vs. $\boldsymbol{M}_{\uparrow\downarrow}$ | MNIST w/ vflip | 250 x 20 | 3 | 0.1 |
| $\boldsymbol{C}_{A-S}$ vs. $\boldsymbol{C}_A$ | CUB | 5000 | 5 | 0.1 |
| $\boldsymbol{C}_{A-YB}$ vs. $\boldsymbol{C}_A$ | CUB | 5000 | 5 | 0.1 |
| $\boldsymbol{C}_{A-YC}$ vs. $\boldsymbol{C}_A$ | CUB | 5000 | 5 | 0.1 |
| $\boldsymbol{C}_{A-E}$ vs. $\boldsymbol{C}_A$ | CUB | 5000 | 5 | 0.1 |
| $\boldsymbol{C}_{A-D}$ vs. $\boldsymbol{C}_A$ | CUB | 5000 | 5 | 0.1 |
| $\boldsymbol{M}_D$ vs. $\boldsymbol{M}_{D2}$ | (P) Primates-[gibbon, siamang, spider monkey] (ImageNet) | 500x3 | 5 | 0.05 |
| $\boldsymbol{M}_D$ vs. $\boldsymbol{M}_{D2}$ | (MT) Mittens-[mitten, Christmas stocking, sock] (ImageNet) | 500x3 | 5 | 0.05 |
| $\boldsymbol{M}_D$ vs. $\boldsymbol{M}_{D2}$ | (B) Buses-[trolley bus, school bus, passenger car] (ImageNet) | 500x3 | 5 | 0.05 |
| $\boldsymbol{M}_D$ vs. $\boldsymbol{M}_{D2}$ | (D) Dogs-[whippet, Saluki, Italian greyhound] (ImageNet) | 500x3 | 5 | 0.05 |
| $\boldsymbol{M}_C$ vs. $\boldsymbol{M}_{CN}$ | (C) Corvids-[Crows, Ravens] (iNaturalist) | 500x2 | 5 | 0.05 |
| $\boldsymbol{M}_C$ vs. $\boldsymbol{M}_{CN}$ | (G) Gators-[American Alligator, American Crocodile] (iNaturalist) | 500x2 | 5 | 0.05 |
| $\boldsymbol{M}_C$ vs. $\boldsymbol{M}_{CN}$ | (MP) Maples-[Sugar Maple, Red Maple, Norway Maple, Silver Maple] (iNaturalist) | 500x4 | 5 | 0.05 |

