# OpenReview forum: "Representational Difference Explanations"
_NeurIPS.cc/2025/Conference — NeurIPS 2025 poster_

### Official Review · Reviewer_mmBM · 2025-06-15

**Clarity:** 3
**Significance:** 3
**Originality:** 3
**Rating:** 4
**Confidence:** 3

**Summary:**

This paper proposes a method called Representational Differences Explanations (RDX), aimed at identifying and explaining differences between the learned representation spaces of two models. The method automatically identifies groups of input samples that are close in one model’s representation space but far apart in the other, or vice versa. The authors introduce a quantitative metric to assess how well such groups reflect the divergence between models’ internal representations.
RDX is compared against classical clustering (k-means) and sparse dictionary learning methods (e.g., NMF, Sparse Autoencoders). The evaluation is carried out on controlled settings such as MNIST and CUB, and the paper includes an ablation study and qualitative examples that demonstrate how RDX can uncover previously unknown model differences.

**Questions:**

(1) Did/Can dictionary learning methods uncover any different types of representational differences compared to RDX?

(2) line 57: The statement “Visual concepts are usually defined as images that the model considers to be similar” may be unprecise. I was under the impression that visual concepts rather correspond to recurring input patterns or localized semantics, not just described by overall image similarity.

**Ethical Concerns:**

["NO or VERY MINOR ethics concerns only"]

**Final Justification:**

The paper introduces RDX, a novel and intuitive framework for identifying and explaining differences between the learned representation spaces of two models. This is a relatively unexplored but important direction in interpretability research.
Strengths include a clear problem formulation, a principled combination of established methods, and a simple yet effective metric for assessing difference explanations. The qualitative case studies convincingly demonstrate RDX’s ability to uncover subtle and non-obvious representational differences.
The rebuttal addressed an earlier concern by providing results with additional baselines, which strengthens the empirical contribution.
However, some issues remain: scalability to large datasets is not addressed (the method is limited by quadratic complexity in the number of samples), and the conditions under which other, more established baselines might perform favorably remain underexplored.
Overall, the novelty of the problem setting, the soundness of the approach, and the improved empirical evidence after the rebuttal outweigh the remaining limitations. I maintain a positive recommendation, with the expectation that future work will explore scaling strategies and broader baseline comparisons.

**Limitations:**

yes

**Paper Formatting Concerns:**

No major formatting issues.

**Quality:**

3

**Strengths And Weaknesses:**

Strenghts:
- The paper tackles the challenge of systematically explaining differences between learned representations across models, a question that has received far less attention than single-model interpretability.
- The paper is clear and accessible. Many natural questions that arise while reading are addressed in the appendix, making the paper thorough without being dense.
- RDX is composed of well-established components, such as spectral clustering and local neighborhood distances, combined in a novel way to address the representation difference problem.
- The authors propose a concrete, intuitive metric to assess the quality of difference explanations. While simple, it aligns well with the goals of the method and is easy to compute.
- The method is evaluated both quantitatively and qualitatively on controlled datasets like MNIST and CUB.
- The case studies clearly show that RDX can uncover subtle, meaningful, and non-obvious differences between model representations.
- The comparison to dictionary learning methods (e.g., NMF, Sparse Autoencoders) illustrates where these classical approaches fall short in identifying contrastive differences between models.

Weaknesses:
- The baselines used in the paper (e.g., k-means, basic sparse dictionary learning) are relatively simple and unsurprisingly underperform. This weakens the strength of the empirical comparisons. Incorporating more recent and relevant methods, such as those explicitly designed to compare representation spaces [1, 2], would provide a stronger benchmark. In particular, USAE [1] could be applied by training a larger sparse autoencoder over both models and then identifying concepts that are selectively activated in only one model.
- The method relies on computing pairwise distance matrices, which scale quadratically with the number of samples. While experiments use up to 5,000 samples, this becomes problematic for large-scale datasets (e.g., millions of samples). The paper does not address how RDX would scale to such settings, where sparsity of concepts or subsampling may become essential.
- The SAE comparison is limited to very low-dimensional hidden spaces (e.g., 3 or 5 units/neurons), which may restrict the richness of the discovered differences. Modern alternatives like top-K sparse autoencoders could offer more expressive and interpretable explanations, and should be considered
- The method assumes that shared local concepts imply similarity in the representation space. However, this assumption may not hold in more complex settings where the relevant concept is a small part of the input (and background features differ a lot). In such cases, using input-space similarity to assess representational similarity might be misleading or insufficient.

References:

[1] Thasarathan, Harrish, et al. "Universal Sparse Autoencoders: Interpretable Cross-Model Concept Alignment." arXiv preprint arXiv:2502.03714 (2025).

[2] Vielhaben, Johanna, et al. "Beyond scalars: Concept-based alignment analysis in vision transformers." arXiv preprint arXiv:2412.06639 (2024).

---

> ### Author Rebuttal · Authors · 2025-07-30
>
> We thank **mmBM** for their comments and suggestions. Based on the requests, we have added new quantitative comparisons to further demonstrate the effectiveness of our RDX approach.
>
> **(mmBM-1) Baselines are relatively simple. Can we compare to more recent baselines?**
> In the paper we compare to a range of distinct baselines which span different families of methods, e.g., PCA, dictionary learning, and sparse autoencoders (see Fig. 3). They have been used with some success in interpretability tasks (e.g., in [D, E]) and we felt it was important to assess them on the model comparison task in this work.
>
> As indicated in the review, there have been some recent contemporaneous works comparing model representations, e.g., [A] on arXiv in Feb 2025,  [B] on arXiv in Dec 2024, and [C] on arXiv in Mar 2025. We already discuss [A, C] in our related works section (see L72 and L70). We found several challenges when trying to fairly compare to USAEs [A], so we decided to exclude it.
> * The official code for USAEs has not yet been published at the time of this rebuttal. The USAE submission on OpenReview has a link to a GitHub page, but this leads to a 404 page. Also, the overcomplete repository (a Vision-based SAE Toolbox on GitHub) appears when searching for the USAE code, but this is because the README states that overcomplete was used by the USAE authors. Despite this, the code for training USAEs was not available in overcomplete.
> * The methodology proposed by the authors of [A] requires training a universal representation space for every comparison. We had concerns about fairly evaluating their methodology due to the number and variety of comparisons in this work. For example, we conduct comparisons for several model variants trained on modified MNIST datasets, PCBM models trained on CUB, models trained on ImageNet, and models trained on iNaturalist. It is not clear if USAEs will be sensitive to dataset size and/or the number of models used in learning the universal latent space.
> * Additionally, we work with small focused datasets with similar classes, it has not been established how to correctly apply USAEs to small scale settings like the modified MNIST experiment we present in Fig. 2.
> * We also would also like to highlight Fig. A2, which shows several issues that are true of any linear decomposition method that make them less ideal for comparison tasks.
> * Finally, RDX is explicitly designed to find differences, whereas both USAEs [A] and RSVC [C] *may* find differences as a byproduct of their main goals.
>
> Thank you for the suggestion to include a comparison to NLMCD [B]. We were able to run experiments with this approach on our unaligned ImageNet representations (i.e., DINO vs DINOv2) and iNaturalist experiments from Fig. 6 and Figs. A7-A10. Note that this is a total of 7 experiments. We had to adapt NLMCD to make it compatible with our evaluation protocol as it generates an arbitrary number of concepts for each representation and measures the similarity of concepts from each representation. For our experiments, we selected the *k* concepts that had the lowest similarity score as the representative concepts for differences and measured the BSR metric. We find that RDX outperforms NLMCD consistently on all 7 experiments. We provide first, second and third quartiles for the BSR metric.
>
> |                        | RDX                  | KMeans           | SAE              | NLMCD            | TopKSAE          |
> |------------------------|----------------------|------------------|------------------|------------------|------------------|
> | Unaligned (Q1, Q2, Q3) | **0.90, 0.94, 0.97** | 0.81, 0.88, 0.92 | 0.54, 0.60, 0.64 | 0.45, 0.52, 0.59 | 0.71, 0.76, 0.82 |
>
> [A] Thasarathan, et al. "Universal Sparse Autoencoders: Interpretable Cross-Model Concept Alignment." arXiv:2502.03714 2025
> [B] Vielhaben, Johanna, et al. "Beyond scalars: Concept-based alignment analysis in vision transformers." arXiv:2412.06639 2024
> [C] Kondapaneni et al... "Representational Similarity via Interpretable Visual Concepts." ICLR 2025
> [D] Fel, et al. "Craft: Concept recursive activation factorization for explainability." CVPR 2023
> [E] Cunningham, et al. "Sparse autoencoders find highly interpretable features in language models." arXiv:2309.08600 2023
>
> **(mmBM-2) Scaling due to requirement for pairwise distance matrices.**
> Our intended setting for this approach is the fine-grained analysis of a smaller dataset. We believe this setting to be quite reasonable for scientists (such as astronomers or biologists) studying specific datasets with vision models. Scaling this approach would require a sub-sampling strategy using a sparse similarity matrix to find groups of images that are near each other in the representational space. Beyond constructing a large similarity matrix, we see no other conceptual issues associated with applying our method to larger datasets.
>
> Additionally, we would like to highlight that it is not clear that analyzing models at scale results in a meaningful understanding of the underlying models, as seen in recent evaluation studies [F, G]. We believe our approach of focusing on smaller datasets makes it easier to assess the efficacy of a method and will be more fruitful in the long-run. However, we agree that this is an important limitation to highlight and we will discuss it in the revised text.
>
> [F] Kantamneni et al. "Are sparse autoencoders useful? a case study in sparse probing." arXiv:2502.16681 2025
> [G] Smith et al. "Negative results for SAEs on downstream tasks and deprioritising SAE research (GDM mech interp team progress update# 2).” 2025
>
> **(mmBM-3) Issue with low dimensional SAEs. Why not use top-K SAEs with a larger latent space?**
> We chose to use low-dimensional SAEs to have a fair, consistent, and straightforward evaluation for all methods. However, it is true that SAEs tend to work better with an overcomplete basis. Based on this  suggestion, we trained a top-K SAE with 50 concepts on unaligned and aligned experiments on the ImageNet subsets and iNaturalist subsets. We selected the 5 concepts with the largest average weight as representative concepts for our evaluation.
>
> We present the BSR results of the 7 “unknown” experiments here for both aligned and unaligned representations. Q1, Q2, and Q3 indicate the first quartile, median, and third quartile respectively.  We found that top-K SAEs outperformed SAEs, but still underperformed RDX, especially when comparing aligned representations. Aligned representations generally have more subtle differences making it more challenging for methods that are not designed to seek out differences.
>
> |                        | RDX                  | KMeans           | SAE              | NLMCD            | TopKSAE          |
> |------------------------|----------------------|------------------|------------------|------------------|------------------|
> | Unaligned (Q1, Q2, Q3) | **0.90, 0.94, 0.97** | 0.81, 0.88, 0.92 | 0.54, 0.60, 0.64 | 0.45, 0.52, 0.59 | 0.71, 0.76, 0.82 |
> | Aligned                | **0.89, 0.91, 0.94** | 0.51, 0.60, 0.69 | 0.38, 0.44, 0.50 |       ---        | 0.35, 0.41, 0.57 |
>
> Also, we are happy to include additional NLMCD results on Aligned representations for the final version. We did not have time to run this experiment as it requires some extra modifications of the NLMCD code.
>
> **(mmBM-4) Assumption that shared local concepts imply similarity in the representation space.**
> The assumption our work makes is actually the inverse statement. We assume that similarity in the representational space implies that there is a shared concept in the input space. This is a commonly used assumption in concept discovery, eg., [D, E]. For example, in Craft [D], concepts are visualized via the maximally activating images for that concept. The images that fit this criteria are considered similar in the representational space, specifically along the direction of that concept.
>
> By visualizing images that are considered similar *by the model*, we hope to identify a consistent visual feature that the model has picked up on. If the model organizes images by a subtle, small concept, then we assume it will be one of the consistent visual attributes in our explanation grids. We will clarify this assumption in the revised text.
>
> **(mmBM-5) Can dictionary learning methods uncover any different types of representational differences compared to RDX?**
> We found it challenging to interpret the results from the dictionary learning methods due to the issues we highlight in Fig. A2, i.e., even when there was a concept discovered by a method, it was unclear how unique it was and how to interpret it alongside the other concepts extracted by the method. In contrast, RDX identifies concepts that maximize the uniqueness of the concept and does not require reasoning over multiple concepts in a weighted linear combination. In Fig. 2, we show how RDX focuses on the core difference between representations and results in easy to interpret explanations.
>
> **(mmBM-6) Definition of visual concepts on L57.**
> We appreciate how this definition could lead to confusion. Generally there are two approaches to concept based XAI methods: (1) concepts are pre-defined via a set of inputs [H] and then the model is analyzed to see how much the concept contributes to decision making or (2) concepts are discovered directly from the model’s activations [D]. Our definition only applies to the second setting, we will adjust the definition to be more general. Thanks for the suggestion.
>
> [H] Kim et al. "Interpretability beyond feature attribution: Quantitative testing with concept activation vectors (tcav)." ICML 2018

---

> > ### Author Response · Authors · 2025-08-06
> >
> > Greetings mmBM,
> >
> > Do not hesitate to let us know if you have any additional questions regarding our response to your review.

---

> > ### Comment · Reviewer_mmBM · 2025-08-07
> >
> > Thank you for the additional experiments. These substantially strengthen both the usefulness and the empirical support for the method.
> >
> > One important point I would encourage the authors to clarify in the paper is the assumption: "We assume that similarity in the representational space implies that there is a shared concept in the input space". This assumption does not necessarily hold in all cases. For example, if images depict various animals in forest environments, the dominant features in the representation space may relate to the animals rather than the shared "forest" background. In such a case, even though the input space shares a common concept, representational similarity may remain low.
> >
> > That said, since this assumption is common in the literature, it would be helpful to both cite relevant precedents and briefly acknowledge its limitations in the paper.
> >
> > After reading the response, I now lean toward increasing my score.

---

> > > ### Author Response · Authors · 2025-08-07
> > >
> > > Indeed, this is a common assumption and we will cite relevant papers (e.g., see [9, 14, 16, 17]) and acknowledge this assumption more clearly in the paper. Thanks for the suggestion.

---

### Official Review · Reviewer_e4CM · 2025-06-30

**Clarity:** 4
**Significance:** 3
**Originality:** 4
**Rating:** 5
**Confidence:** 4

**Summary:**

The paper proposes a new method, Representational Differences Explanations (RDX), to compare representations of different models. The authors evaluate the method both on representations with known and unknown differences and show examples where RDX yields better explanations of the differences.

**Questions:**

(see the weaknesses for more information and context)
1) Can you provide any guarantees that changing the dataset would have only a minor effect on the final explanations? Or, at least, show it empirically?
2) Is there any reason to believe that using Euclidean distances in your method makes sense even when the data lies on curved manifolds?
3) How did you choose the examples of explanations shown in the paper? Were there any objective criteria used for choosing what to report (and what not to)?
4) Regarding representational alignment (section 3.4): it is not clear to me when exactly this step happens. Do you use the transformation in all the experiments, or only in some? How does it change the resulting explanations?

**Ethical Concerns:**

["NO or VERY MINOR ethics concerns only"]

**Final Justification:**

I was satisfied with the responses to my comments, and I also did not find anything disturbing in the other reviewers. Hence, I raised the score.

**Limitations:**

Yes

**Paper Formatting Concerns:**

No issues

**Quality:**

3

**Strengths And Weaknesses:**

Overall, I think the paper proposes an interesting method that can give new insights into the differences between the representations of models. However, there are some problems that need to be addressed or justified.

**Strengths:**
- The problem, as described in the Introduction and illustrated by Figure 1, is well-motivated and easy to understand.
- The paper is clear, well-written, and easy to understand.
- The problem addressed in this paper is timely and important.
- It seems to me that there are sufficient details needed to replicate the experiments.
- Based on the figures provided, it seems that the method works and can reveal interesting insights into the differences between the models. It is able to capture differences between two models instead of focusing on the most important explanations for each model separately.

**Weaknesses:**
- The examples shown might potentially be cherry-picked. For example, in Figure 6, five different explanations were identified in each case, but images are shown only for two of them. How can one make sure that these were not shown solely because they support the claims and make a nice story?
- Thorough evaluation of the method is missing. The binary success rate metric (BSR) seems to me a bit cyclic, since it evaluates something that the RDX method was explicitely developed to do (identify pairs that are closer in one representation than the other). But the other methods, that RDX are compared to, do not have this as a primary goal, so it is perhaps not so surprising that RDX outperformes the rest. But does it neccessarily imply that it gives better explanations? (I also acknowledge that objectively evaluating explanations is a complicated issue that perhaps goes beyond the scope of this paper.)
- The method depends on Euclidean distances between points from a chosen dataset. I think this is problematic from two perspectives. First, there is a strong dependency on the dataset choice. Since the distances are defined as a ranking of the nearest neighbors (and not how far the points actually are), adding or removing points from the dataset would heavily influence the distance matrices. For instance, in the experiments where only a subset of ImageNet classes are used, would the explanations be different if all ImageNet classes were used for testing? Second, the representations usually live on curved manifolds in high-dimensional spaces, where Euclidean distances do not make sense. For example, two points can be close in Euclidean geometry, but very distant in manifold geometry. How are these differences taken into account in RDX?

---

> ### Author Rebuttal · Authors · 2025-07-30
>
> We thank **e4CM** for their comments and suggestions. Please do not hesitate to follow up in the discussion period if there are any remaining questions or points that we can clarify.
>
> **(e4CM-1) How were the qualitative results in the paper selected?**
> We select the clearest and most representative examples for the main paper. We include the full set of explanations for most comparisons in the appendix. We selected two concepts to visualize in Fig. 6 due to space constraints, the remainder are available in Figs. A8 and A10. More results for MNIST are found in Fig. A4 and more results for the CUB experiments are shown in Figs. A5 and A6. Complete results for the Mittens (ImageNet), Primates (ImageNet), Maples (iNat), and Corvids (iNat) are available in Figs. A7-A10. There is no specific objective criteria for choosing which results are presented in the main text vs. presented in the appendix. However, we feel that we have presented a large number of results from our experiments for the reader to inspect.
>
> **(e4CM-2) BSR is cyclic. Is it possible to improve evaluation?**
> We designed BSR to measure a property of difference explanations that we believe is important for comparing representations, i.e., finding groups of inputs that are considered similar by only one of two representations. We show that existing methods do not succeed at this task and design a method, RDX, to optimize for this metric (see Figs. 2, 4, and 5). We agree that it is perhaps not surprising that our method performs better than existing methods for this task, but we report BSR as it demonstrates that we have succeeded in our goals. Additionally, we have tested BSR with different distance normalization schemes and found that RDX generally performs the best even when the distance normalization schemes are mismatched (Fig. A13).
>
> We agree that an in-depth human evaluation is out of the scope of this work. We include many qualitative visualizations in both the main paper and appendix for the reader to scrutinize and assess if the explanations are meaningful. In addition, in Sec. B.3 we report results for some case studies to discuss how explanations may, or may not, link to classification performance.
>
> To strengthen our quantitative results further, following suggestions from **jwKi** and **mmBM**, we include new metrics from [A] and a new method that was explicitly designed for comparison [B] in our results. The new metrics can be found in the response to **jwKi** (see response **(jwKi-1)**).  The new method’s results can be found in the response to **mmBM** (see response **(mmBM-1)**).
>
> [A] Dreyer, et al. "Mechanistic understanding and validation of large AI models with SemanticLens." arXiv:2501.05398 2025
> [B] Vielhaben, et al. "Beyond scalars: Concept-based alignment analysis in vision transformers." arXiv:2412.06639 2024
>
>
> **(e4CM-3) Points may be nearby using Euclidean distance, but far apart on the data manifold. How is this handled?**
> This is a good point, two datapoints may indeed be close together based on Euclidean distance but quite distant if we follow the manifold that the data lies on. We do not explicitly handle this case, but there are two reasons we believe RDX can work despite this issue.
>
> (1) Many models (e.g., like DINO) are trained to have meaningful Euclidean distances in their latent space and so we assume the distances are meaningful. For classification models, the last layer is a linear classifier, resulting in a linearly separable representation. While this does not guarantee meaningful Euclidean distances, it strongly implies it. We use the last layer for all experiments in this work.
>
> (2) It is unlikely that the *nearest* neighbors for a given point have small Euclidean distances, but large distances on the manifold, since we generally expect that representations are *locally* linear. Neighborhood distances and our locally-biased difference function push RDX to seek out groups of points that are locally linear. Notably, the non-linear dimensionality reduction technique Isomap [C] uses top-k nearest neighbors to construct a graph and estimate distances along the manifold of the data. Our locally-biased difference function is a *soft* version of the same idea.
>
> Our results are reported across a variety of backbones and training objectives (e.g., see Table A5) and demonstrate robustness in the presence of this variation.  Finally, RDX is not restricted to Euclidean distances. In the future, it would be interesting to explore other choices for the various components of the algorithm.
>
>
> [C] Tenenbaum, et al.. "A global geometric framework for nonlinear dimensionality reduction." Science 2000
>
>
> **(e4CM-4) How does adding or removing points affect explanations?**
> RDX explanations are conditioned upon both the model and the input data. To the best of our knowledge, this is true for all concept-based explanation methods. For example, dictionary-learning approaches (NMF, SAE, PCA, etc…) will have different dictionaries depending on the data used to generate model activations. If the input data changes, the representation will change, and, subsequently the explanations will change.
>
> However, if the number of points changes, but the data distribution stays the same, the concepts discovered will be very similar. We have conducted a new experiment, in which we add or remove 20% of the data (from the same distribution) and observed that the spectral clusters are highly similar. We are happy to include this result in the revised paper. Thanks for the suggestion!
>
>
> **(e4CM-5) How does representational alignment work? When does this step happen, how does it change the explanations?**
> The step does not happen for all experiments. For the “unknown” difference experiments (ImageNet and iNaturalist), we conduct experiments without and with alignment and report results for both unaligned (Fig. 3B Unaligned) and aligned representations (Fig. 3B Aligned). When aligning representations, we test alignment in both directions. We align representation A to representation B and vice versa. We use the symbol ‘ to indicate aligned representations (see L180). In Fig. 3B Aligned, we can see that results are reported twice for the baseline methods. KMeansA’ indicates that A was aligned to B, and KMeansB’ indicates that B was aligned to A. The qualitative results we show for these unknown experiments are all after alignment (Fig. 6 and Figs. A7-A10).
>
> Representations may be misaligned due to a linear transformation of the representations. In these cases, aligning the representations before comparing them can help us isolate more fundamental differences (Sec 3.4). Alignment does change the explanations. In Fig. A3, we visualize the effect of alignment and show how it changes the explanations. We will highlight this result in the revised text and update the description of the alignment so that it is clearer.

---

> > ### Comment · Reviewer_e4CM · 2025-08-01
> >
> > I would like to thank the authors for their thorough rebuttal and new evaluations. I am satisfied with the answers provided, and I changed the score accordingly. There are still points for further discussion, but that would not have a strong effect on my final decision.

---

### Official Review · Reviewer_jwKi · 2025-06-30

**Clarity:** 4
**Significance:** 4
**Originality:** 4
**Rating:** 5
**Confidence:** 4

**Summary:**

The paper proposes RDX, a novel explanation method that identifies differences in representations between two models and visualizes them as concepts via examples. Explanations are evaluated qualitatively, as well as quantitatively through BSR, a newly proposed metric for evaluating difference explanations.

**Questions:**

- Please rebut my concerns regarding using only the BSR metric for quantitative comparisons or include a comparison via other evaluation metrics, to offer a more holistic view.
- Please answer my above question regarding scalability of RDX beyond pair-wise comparisons

**Ethical Concerns:**

["NO or VERY MINOR ethics concerns only"]

**Final Justification:**

I am satisfied with the author's answers to my questions; however I still believe that evaluation is a slight issue here, due to the unavailability of unbiased evaluation metric(s) for difference explanations (owed in part to the fact that, afaik, the concept of difference explanations is novel, and thus lacks evaluation availability).
Taking this and the other reviews and corresponding rebuttals into account, I believe this paper is a strong 5, and am thus keeping my initial rating.

**Limitations:**

yes

**Quality:**

3

**Strengths And Weaknesses:**

**Strengths.** The paper is extremely well-written and structured. The proposed method is innovative and provides a new type of difference-based concept explanations that will be of significant interest to the XAI community. The addressed issue of revealing behavioral differences between models is explained in detail and supported by experimental results. Explanations are evaluated both qualitatively and quantitatively. The proposed method is compared to several state-of-the-art related methods for dictionary learning. Visualizations are informative. The large appendix ensures reproducibility and provides several additional results that answered many of my initial questions.

**Weaknesses**
BSR, the proposed metric for evaluating explanations (and similarly its variants in the Appendix) simply counts the frequence where two examples from the explanation grid have a lower representational distance in A than in B (i.e., $D_A$ < $D_B$). The sampling of difference explanation grids for RDX is done in a way that only indices with pairwise negative entries in $G_{A,B}$ (line 149), i.e. with $D_A < D_B$ according to Equation (1) are chosen. While I agree that BSR may be a desirable property for difference explanations, its formulation is too close to that of RDX to serve as the (only) quantitative comparison to other explanation methods. I would have liked some comparison via other quantitative metrics as well, e.g., evaluating how/if the found representational differences affect model performance or some of the metrics listed here [1] to offer a more holistic comparison to other explanation methods.

IIUC, since RDX is not symmetric, comparing two models requires inspecting two sets of explanations: RDX(A, B) and RDX(B, A) to discover potential representational differences. I am wondering how and if this scales to comparing more than two models at once, due to this asymmetry.

[1] Dreyer, Maximilian, et al. "Mechanistic understanding and validation of large AI models with SemanticLens." arXiv preprint arXiv:2501.05398 (2025).

---

> ### Author Rebuttal · Authors · 2025-07-30
>
> We thank **jwKi** for their careful reading of the paper and insightful comments and suggestions.
>
>
> **(jwKi-1) Concerns about using only the BSR metric.**
> We agree that BSR is a desirable property for difference explanations and for this reason our design of RDX is optimized for the BSR target metric. We report performance using this metric as it best matches our methodological goals, i.e., finding groups of inputs that are considered similar by only one of two representations. Additionally, we report results for the BSR metric using alternative distance normalization methods in Fig. A13. There we observe that even when the normalization method used by RDX and the normalization method used in evaluation with BSR are mismatched, RDX still performs the best. This demonstrates RDX’s robustness even when not aligned directly with the evaluation metric.
>
> Nonetheless, we agree that additional evaluation metrics would strengthen our results further. To address this we report additional results using the metrics from Dreyer, et al. [A] as suggested. These metrics use an independent, “semantic” model, OpenCLIP, to assess the *clarity*, *redundancy*, and *polysemanticity* of a concept via its explanation. We computed these metrics for the ImageNet and iNaturalist experiments (from Sec. 4.4 and Fig. 6) for both unaligned and aligned representations. Note that for *redundancy*, we adapt the metrics to compare the explanation grids from each model. Essentially, we compute the *similarity* for each concept from Model A to each concept from Model B. We use this similarity matrix to compute the *redundancy* metric (which summarizes the similarity matrix). We present the results below. Q1, Q2, and Q3 indicate first quartile, median, and third quartile, respectively.
>
> **Unaligned**
> |                      | RDX                  | KMeans               | SAE              | CNMF             | PCA              |
> |----------------------|----------------------|----------------------|------------------|------------------|------------------|
> | Clarity (Q1, Q2, Q3) | 0.38, 0.43, 0.49     | **0.41, 0.45, 0.51** | 0.35, 0.38, 0.42 | 0.39, 0.44, 0.49 | 0.39, 0.43, 0.51 |
> | PolySem              | 0.29, 0.36, 0.44     | **0.27, 0.33, 0.41** | 0.35, 0.41, 0.47 | 0.29, 0.35, 0.42 | 0.31, 0.35, 0.45 |
> | Redundancy           | **0.76, 0.80, 0.82** | 0.83, 0.86, 0.87     | 0.81, 0.84, 0.85 | 0.81, 0.84, 0.85 | 0.76, 0.84, 0.85 |
>
> The reduction in redundancy is even more apparent after alignment.
>
>
> **Aligned**
>
> |                      | RDX                  | KMeans               | SAE              | CNMF                 |
> |----------------------|----------------------|----------------------|------------------|----------------------|
> | Clarity (Q1, Q2, Q3) | 0.44, 0.45, 0.46     | 0.45, 0.45, 0.49     | 0.42, 0.43, 0.49 | **0.48, 0.52, 0.57** |
> | PolySem              | 0.28, 0.32, 0.45     | **0.26, 0.32, 0.37** | 0.40 0.43, 0.44  | 0.34, 0.34, 0.35     |
> | Redundancy           | **0.76, 0.78, 0.79** | 0.85, 0.87, 0.90     | 0.81, 0.81, 0.82 | 0.82, 0.84, 0.86     |
>
> We observe that RDX generates significantly less redundant explanations as we would expect. We do notice that the RDX explanations are slightly less clear than some other baseline methods according to some of these metrics. However, most methods have similar clarity and polysemanticity scores. We would like to emphasize three caveats to interpreting these results: (1) RDX generates explanations for differences, which naturally highlights parts of the representations that are more likely to be complex and less clear. (2) The metrics here presume that the generalist model (OpenCLIP) is an unbiased evaluator, but it is likely that OpenCLIP has its own biases that influences what it estimates is more or less clear/redundant. (3) Explanations for complex, niche datasets that are out of domain for OpenCLIP would likely have significant issues when evaluated using this metric. A benefit of the BSR metric is that it is insensitive to the complexity of the input images being analyzed.
>
> Additionally, RDX is the only method in this work that is able to isolate and explain differences between two representations as seen in Figs. 2, 4, 5, 6, and A4-A10. We believe that a slight increase in concept complexity is natural and acceptable when focusing on understanding nuanced representational differences.
>
> [A] Dreyer, et al. "Mechanistic understanding and validation of large AI models with SemanticLens." arXiv:2501.05398 2025
>
>
> **(jwKi-2) Scalability of RDX beyond pairwise comparisons.**
> This is an interesting question. There are at least two challenges that would need to be addressed. Suppose we have three representations, A, B, and C. (1) If we want to identify the differences between A vs. “B and C”, we have to decide if this means the union of the differences between (A, B) and (A, C), or the intersection. (2) We would need to develop a method to merge difference graphs from each pairwise comparison so that the output of clustering is the union/intersection of the concept sets. While this would scale combinatorially, RDX is fast enough on datasets of 5k points with a 512 dimensional latent space (32 seconds), that in some settings the pairwise comparisons would be feasible. Our work represents one of the first approaches to automatically identify explainable differences between representations from two different models. Extending our work efficiently beyond pairs of models, is a really interesting question which we leave for future work. We will highlight this opportunity in the revised text.

---

> > ### Comment · Reviewer_jwKi · 2025-08-04
> >
> > Thank you for the thorough rebuttal. I am happy with the answers and have finalized my score accordingly. While I have some follow up discussion points, they would not affect my decision.

---

### Official Review · Reviewer_HVHo · 2025-07-02

**Clarity:** 2
**Significance:** 3
**Originality:** 3
**Rating:** 4
**Confidence:** 2

**Summary:**

The paper introduces Representational Difference Explanations (RDX), a post-hoc analytic technique that surfaces where two models disagree in their learned representations. RDX builds pairwise distance matrices for two embedding spaces, forms an asymmetric, locally-biased difference matrix that emphasises disagreements on near-neighbours, converts that matrix into an affinity graph and applies spectral clustering, then turns each high-affinity cluster into a concise image grid that itself is the explanation. Experiments showcase the metholodogies performance on CUB, MNIST and large-scale “unknown difference” settings.

**Questions:**

Questions:
- Could the authors explain their choice for conducting experiments on a modified MNIST setting, instead of the original?
- Is there any comparison with recent explanation methods?
- Could the authors cite some metrics or numerical evaluation for their method is the main text of the paper?

**Ethical Concerns:**

["NO or VERY MINOR ethics concerns only"]

**Final Justification:**

I have read the rebuttal and I am satisfied with the responses to my comments.

**Limitations:**

Yes.

**Paper Formatting Concerns:**

I did not observe any significant deviations from the NeurIPS 2025 formatting guidelines.

**Quality:**

3

**Strengths And Weaknesses:**

Strengths:
- The paper tackles an interesting problem of comparing two models' explanations through post-hoc analysis
- The paper does not require gradient access or optimization, making it practical for real world applications

Weaknesses:
- In sections 4.1 and 4.2, the author conduct experiments for dictionary learning a modified MNIST setting, without explaining the rationale behind this choice. MNIST is utilized for comparing digits as some of them can look similar to others.
-Authors do not compare their method with other XAI methods. The baselines include models that have been largely explored in the literature.
- The evaluation basis of the method is largely visual. Tables are placed in the appendix, making the main text of the paper difficult to follow, as it's very notation-heavy.

---

> ### Author Rebuttal · Authors · 2025-07-30
>
> We thank **HVHo** for their feedback and suggestions. We believe that addressing these comments will make the paper even clearer.
>
> **(HVHo-1) Why conduct experiments on a modified MNIST?**
> In our first experiment, we compare two checkpoints trained on a MNIST dataset modified to include only the digits: 3, 5, and 8 (see L225). Our motivation was to create a simple and clean experimental test bed that could demonstrate an issue we observe with existing explanation methods like NMF, KMeans, PCA, SAE, etc. In Fig. 2, we show that existing methods, that are designed to analyze one representation at a time, are not suitable for comparing representations across models as they result in explanations that are unrelated to the core difference between the representations. In Fig. A2, we discuss issues with explanations that rely on linear decompositions of activation matrices as well. Importantly, we do conduct experiments on more realistic settings using modern models (DINO, DINOv2, CLIP, PCBMs) on real datasets (CUB, ImageNet, and iNaturalist). We will clarify the motivation why we selected this modified version of MNIST in the revised text.
>
> **(HVHo-2) Comparison with additional XAI methods.**
> We compare RDX to a variety of XAI methods (e.g., NMF, KMeans, PCA, and SAEs ) throughout the work (e.g., see Fig. 3). They are popular [A, B] both in practical applications and as the basis of recent research [D, E].  Following the suggestion from **mmBM**, we include comparisons to an additional explanation method (NLMCD) [C], designed for comparing representations specifically, and observe that our method outperforms it on the BSR metric. We test NLMCD on our unaligned representational comparisons for ImageNet and iNaturalist (Fig. 3B, Unaligned). The results are presented below.
>
> |                        | RDX                  | KMeans           | SAE              | NLMCD            | TopKSAE          |
> |------------------------|----------------------|------------------|------------------|------------------|------------------|
> | Unaligned (Q1, Q2, Q3) | **0.90, 0.94, 0.97** | 0.81, 0.88, 0.92 | 0.54, 0.60, 0.64 | 0.45, 0.52, 0.59 | 0.71, 0.76, 0.82 |
>
> [A] Fel, et al. "Craft: Concept recursive activation factorization for explainability." CVPR 2023
> [B] Cunningham, et al. "Sparse autoencoders find highly interpretable features in language models." arXiv:2309.08600 2023
> [C] Vielhaben, et al. "Beyond scalars: Concept-based alignment analysis in vision transformers." arXiv:2412.06639 2024
> [D] Bussmann, Bart, Patrick Leask, and Neel Nanda. "Batchtopk sparse autoencoders." arXiv preprint arXiv:2412.06410 2024)
> [E] Gao, Leo, et al. "Scaling and evaluating sparse autoencoders." arXiv preprint arXiv:2406.04093 2024
>
> **(HVHo-3) Analysis is largely visual, is there numerical evaluation?**
> In our opinion, the most effective way of explaining the behavior of interpretability methods for vision models to humans is via visual examples. Thus, we have used visual examples throughout the paper to demonstrate our results to the reader. However, we also report comparisons to existing methods using *quantitative metrics*. The primary goal of our work is to develop a method that can compare two models and reveal the concepts unique to each one. We measure this quantitatively using the Binary Success Rate (BSR) metric, described in Sec. 3.5. The results of this metric are presented in Fig. 3, in which we compare several methods on comparing models on multiple datasets. Ablation experiments can be found in Table A3. Additionally, following the suggestion from reviewer **jwKi**, we have included new quantitative metrics from [F] and find that RDX performs well under these metrics as well. The table and analysis is available in the response **jwKi-1**.
>
> [F] Dreyer, et al. "Mechanistic understanding and validation of large AI models with SemanticLens." arXiv:2501.05398 2025
>
> **(HVHo-4) Tables are placed in the appendix making it hard to follow due to reliance on notation.**
> Thanks for the suggestion. We will polish the notation and clearly point to the additional results in the appendix.

---

> > ### Author Response · Authors · 2025-08-06
> >
> > Greetings HVHo,
> >
> > Do not hesitate to let us know if you have any additional questions regarding our response to your review.

---

> > > ### Comment · Reviewer_HVHo · 2025-08-08
> > >
> > > Thank you to the authors for the response to my comments. I am satisfied with the responses and I will raise my score.

---

### Decision · Program_Chairs · 2025-09-17

**Decision:**

Accept (poster)

**Comment:**

This submission addresses the problem of discovering and explaining differences in the learned representations of two models, proposing a method called Representational Differences Explanation (RDX) as well as a related Binary Success Rate (BSR) evaluation metric.

Strengths identified by the reviewers are as follows:
- The problem is novel and important (Reviewer jwKi calls it a "new type of difference-based concept explanations that will be of significant interest to the XAI community").
- Both qualitative and quantitative evaluation:
    - The qualitative examples show convincingly that RDX can reveal non-obvious differences.
    - The quantitative evaluation uses the proposed BSR metric, which is a contribution in itself.
- The paper is well-written and organized, and includes a long appendix that helps with reproducibility and anticipates reader questions.

Weaknesses:
- Reviewers commented that the BSR metric is closely related to the RDX method and thus other quantitative metrics should be considered. The author rebuttal reported additional metrics from a paper suggested by Reviewer jwKi, and explained why the new results are sensible. Reviewers did acknowledge that the lack of evaluation metrics is understandable given the newness of the problem.
- Reviewers also noted that the baseline methods are not explicitly designed to compare representations. The rebuttal addressed this by reporting on additional comparisons with NLMCD by Vielhaben et al. (2024) and also with top-K SAEs. It also explained why comparison with USAEs is difficult (and USAEs should be treated as contemporaneous).
- RDX has some shortcomings, which could be clarified as limitations:
    - Quadratic scaling with the number of samples.
    - Dependence on a chosen dataset. This is also true of dictionary learning approaches, and the rebuttal promised to report on a new experiment where 20% of points are removed/added, which the authors should follow through on.
    - Dependence on Euclidean distance, although the rebuttal discussed mitigating factors.

All reviewers were in favor of acceptance, and the rebuttal strengthened the quantitative evaluation by adding additional metrics and baselines, which should be incorporated. I also recommend acceptance and feel that this was a well-executed piece of work.